# PHYSICS-ENHANCED NEURAL OPERATOR: AN APPLICATION IN SIMULATING TURBULENT TRANSPORT

## ABSTRACT

Accurate simulation of turbulent flows is of immense importance in a variety of scientific and engineering fields, including climate science, freshwater science, and the development of energy-efficient manufacturing processes. Within the realm of turbulent flow simulation, direct numerical simulation (DNS) is widely considered to be the most reliable approach, but it is prohibitively expensive and thus has limited applicability to long-term and fine-scale simulation over various configurations. Given the pressing need for efficient simulation, there is an increasing interest in building machine learning models for simulating turbulence, either by reconstructing DNS from alternative low-fidelity simulations or sequentially predicting DNS from historical data. However, conventional machine learning models are not designed for capturing complex spatio-temporal characteristics of turbulent flows. This results in their limited performance and generalizability, especially when applied to complex flow data and different flow configurations. This paper presents a novel physics-enhanced neural operator (PENO) that efficiently models the complex flow dynamics while leveraging physical knowledge of partial differential equations (PDEs) to enhance simulation process. we further introduce a self-augmentation mechanism to reduce the accumulated errors in long-term simulations. The proposed method is evaluated through its performance on multiple turbulent flow datasets, showcasing the model's capability to reconstruct high-resolution DNS data, maintain the inherent physical properties of flow transport, and transfer across various resolution settings and simulation configurations. These encouraging results confirms its applicability to a wide range of real-world scenarios in which extensive simulations are needed under diverse settings.

## 1 INTRODUCTION

Advances in computational fluid dynamics (CFD) have significantly impacted various scientific and engineering domains. In the clean energy sector, CFD is essential for enhancing power generation and distribution, including the design of high-efficiency wind turbines and their strategic positioning to maximize energy capture. In the aerospace industry, CFD plays a critical role in analyzing aerodynamic forces and thermal effects on aircraft, rockets, and spacecraft. In particular, efficient CFD techniques are crucial for modeling and refining airflow around wings, fuselages, and engine components, which can help improve fuel efficiency, reduce drag, and enhance maneuverability and safety. Furthermore, CFD is indispensable in climate science for predicting pollution patterns, enhancing emission controls, and assessing the environmental impact of infrastructure projects.

In the field of computational fluid dynamics (CFD), the simulation of turbulent flows, particularly at high spatial resolutions and over long periods, is a crucial task for many scientific applications. Direct numerical simulation (DNS) is widely considered to be the most reliable approach to produce detailed turbulence simulations of high fidelity. However, DNS requires substantial computational resources, which limits its practicality for long-term simulations at fine spatial scales (Givi, 1994).

There are two common approaches to address this issue using data-driven methods. The first approach aims to reconstruct DNS data from the low-fidelity large eddy simulation (LES) (Fukami et al., 2019; 2021; Liu et al., 2020; Xu et al., 2023; Yang et al., 2023). Specifically, LES filters out the smaller scales of turbulent transport (Sagaut, 2005), and consequently, it only generates low-

fidelity simulations on coarser grids (Nouri et al., 2017). Most of these approaches are based on super-resolution (SR) techniques (Park et al., 2003), which have been highly successful in generating high-resolution data in various commercial applications. Predominantly, SR models employ convolutional network layers (CNNs) (Albawi et al., 2017) to identify and transform spatial features into high-resolution images through non-linear mappings. From the initial end-to-end SRCNN model (Dong et al., 2014), researchers have utilized additional structural elements, including skip-connections (Zhang et al., 2018b; Ahn et al., 2018; Dai et al., 2019; Van Duong et al., 2021), channel attention (Zhang et al., 2018a), adversarial training objectives (Ledig et al., 2017; Wang et al., 2018a;b; Karras et al., 2017; Upadhyay & Awate, 2019; Cheng et al., 2021; Wenlong et al., 2021), and more recently, Transformer-based structures (Fang et al., 2022a; Lu et al., 2022; Fang et al., 2022b; Wang et al., 2022; Zou et al., 2022; Liang et al., 2022), and the implicit neural representation methods (Chen et al., 2022). Despite their popularity, these methods remain limited in their accuracy for reconstructing detailed flow patterns, which is primarily due to the lack of physical information about the small-scale flow transport in the low-resolution LES data.

To retain detailed turbulence patterns and capture temporal dynamics in turbulence flows, the sequential prediction method has been developed for generating high-resolution DNS data directly from historical high-resolution DNS data. Specifically, the sequential prediction method employs temporal modeling structures to capture underlying flow dynamics, which can be further enhanced by integrating governing partial differential equations (PDEs) (Omori & Kotera, 2007). This can be achieved by incorporating PDEs into the neural network's learning process (Cai et al., 2021; Eivazi et al., 2022; Kag et al., 2022; Yousif et al., 2022) or by directly encoding PDEs within a recurrent unit (Bao et al., 2022; Chen et al., 2023). Recently, neural operator-based methods (Lu et al., 2019; Li et al., 2020; Wen et al., 2022; Equer et al., 2023; Boussif et al., 2022) have also shown promise in sequential prediction for the Navier-Stokes equation (Foias et al., 2001). The main advantage of neural operator-based methods is their generalizability to different boundary and initial conditions, and their efficiency in generating simulations. Among these methods, the Fourier neural operator (FNO) (Li et al., 2020) also allows the generation of DNS at higher resolutions in a zero-shot fashion, reducing the need for costly high-resolution training data. However, these approaches are not designed to explicitly leverage the knowledge of PDEs, leading to two major drawbacks. First, it is challenging for these approaches to capture complex flow dynamics, especially when training data are scarce. This becomes a critical issue in the context of complex 3D flows, where these methods often exhibit degraded performance. Their prediction errors also accumulate quickly for continuously modeling complex flows over long periods. Second, they remain limited in generalizing to a heterogeneous set of flow datasets governed by different PDE settings, which if often needed for many manufacturing tasks. In the absence of underlying physics, the model is unable to fully distinguish between different flow behaviors. Even though the model could be fine-tuned towards each new flow dataset, it requires additional cost to generate initial simulations needed for fine-tuning.

In this paper, we propose a novel method, physics-enhanced neural operator (PENO), for enhancing the simulation of turbulent transport over long periods and different flow datasets. This proposed method incorporates the physical knowledge of PDE into the FNO (Li et al., 2020) to effectively model turbulence dynamics and also introduces a new self-augmentation mechanism to mitigate the accumulated errors in long-term simulation. In particular, we complement the Fourier layers in FNO with an additional network branch, which gradually estimates the temporal gradient of target flow variables following the underlying PDE. This combined model structure leverages the physical knowledge to better capture complex flow dynamics even in 3D space while also keeping the power and efficiency of data-driven FNO. Next, we identify a key limitation of the standard Fourier layers in preserving informative high-frequency signals, which degrades the performance in long-term simulation. Hence, we augment the input data at each time through zero-shot super-resolution and random perturbation. By introducing additional high-frequency signals at each time step, this self-augmentation mechanism can help prevent Fourier layers from filtering out important high-frequency information during long-term simulation.

The PENO method has undergone thorough assessments using two sets of data, (i) modeling complex flow dynamics on 3D turbulence data, and (ii) generalization over different flow datasets. For test (i), we utilize two datasets: the forced isotropic turbulent (FIT) flow (Minping et al., 2012), and the Taylor-Green vortex (TGV) flow (Brachet et al., 1984). These assessments demonstrate the PENO's consistent ability to reconstruct data effectively over time and across different resolutions. The effectiveness of each component in the proposed method has been highlighted through both

qualitative and quantitative analyses. For test (ii), we conduct experiments on multiple 2D turbulent flow series to confirm the PENO method's transferability and generalizability. Our implementation is publicly available[1].

# 2 PROBLEM DEFINITION AND PRELIMINARIES

## 2.1 PROBLEM DEFINITION

This study focuses on the transport of unsteady turbulent flows. In every scenario, the flow is treated as Newtonian and incompressible, characterized by a uniform density. Spatially, the coordinates are denoted by the vector $\mathbf{x} \equiv \{x, y, z\}$ in a 3D space or $\mathbf{x} \equiv \{x, y\}$ in a 2D space, while time is indicated by $t$. We denote by $\mathbf{Q}(\mathbf{x}, t)$ the target flow variables (e.g., velocity or vorticity) to be simulated. The pressure, density, and dynamic viscosity in the flow are expressed as $p(\mathbf{x}, t)$, $\rho(\mathbf{x}, t)$, and $\nu$, respectively. As a neural operator-based approach, the proposed PENO aims to create mappings between infinite-dimensional functional spaces, and it treats solutions to PDEs as functions rather than discrete sets of points. Henceforth, we represent flow variables, pressure, and density as time-dependent functions $\mathbf{Q}(t)$, $p(t)$, and $\rho(t)$, respectively, in describing PENO, without explicitly showing the spatial coordinates $\mathbf{x}$.

During the training process, the provided DNS data are obtained at regular time intervals $\delta$, denoted as $\mathbf{Q} = \{\mathbf{Q}(t)\}$, where $t$ belongs to the time range $\{t_0, t_0 + \delta, \ldots, t_0 + K\delta\}$. The goal is to forecast high-resolution DNS data for future time points, specifically at $\{t_0 + (K+1)\delta, \ldots, t_0 + M\delta\}$. Additionally, we have access to the large eddy simulation (LES) data at lower resolutions, represented as $\mathbf{Q}^l = \mathbf{Q}^l(t)$ for $t \in [t_0, t_0 + M\delta]$. Since LES data require less computational effort to generate, we assume that they are available for both training and testing phases.

## 2.2 FOURIER NEURAL OPERATOR

Fourier neural operator (FNO) is designed to approximate the PDE solutions through a transformation in a Fourier space (Li et al., 2020). The intuition is to approximate the Green functions by kernels, which are parameterized by neural networks in the Fourier space. In the simulation of turbulent transport, the FNO approach initially transforms the input $\mathbf{Q}(t)$ at time $t$ from the spatial domain to the frequency domain using the Fourier transformation $\mathcal{F}$, as: $\mathbf{Q}_{\text{in}}(\omega) = \mathcal{F}\{\mathbf{Q}(t)\} = \int_{-\infty}^{\infty} \mathbf{Q}(t)e^{-i2\pi\omega t}dt$, where $\omega$ and $\mathbf{Q}_{\text{in}}(\omega)$ denotes the frequency variables and the Fourier transform of $\mathbf{Q}(t)$, respectively. This process $\mathcal{F}$ allows for capturing the global information of input $\mathbf{Q}(t)$ effectively. A neural network $\mathcal{G}$ then learns the mapping between the Fourier

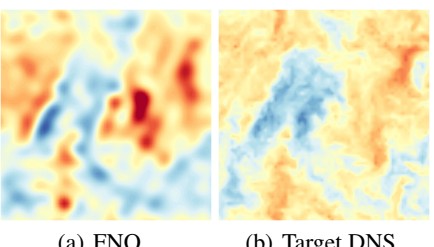

(a) FNO  (b) Target DNS

Figure 1: FNO's prediction vs. true DNS data in the $w$ velocity channel of the forced isotropic flow. This result corresponds to a test point 0.04s (with a sampling interval of 0.002s) following the training period.

coefficients of the input $\mathbf{Q}_{\text{in}}(\omega)$ and output $\mathbf{Q}_{\text{out}}(\omega)$ representation in the frequency domain, essentially approximating the operator of the PDE, as $\mathbf{Q}_{\text{out}}(\omega) = \mathcal{G}(\mathbf{Q}_{\text{in}}(\omega); \phi)$, where $\phi$ represents the parameters of the neural network $\mathcal{G}$. Next, the inverse Fourier transform is applied to convert the learned representation back to the spatial domain, yielding the approximated solution of PDE $\hat{\mathbf{Q}}_{\text{FNO}}(t + \delta)$ at time $t + \delta$, expressed as: $\hat{\mathbf{Q}}_{\text{FNO}}(t + \delta) = \mathcal{F}^{-1}\{\mathbf{Q}_{\text{out}}(\omega)\} = \int_{-\infty}^{\infty} \mathbf{Q}_{\text{out}}(\omega)e^{i2\pi\omega t}d\omega$.

Based on such design, FNO can combine the global information of the entire field embedded through the Fourier transformation and the expressive power of neural networks, enabling the learning and approximation of high-dimensional and complex PDE operators directly from data. Despite the promise of this approach, FNO has several limitations, especially when used in turbulence simulation. Firstly, FNO learns PDEs (e.g., Navier-Stokes equation) from data without knowing the PDEs' format. Hence, it requires a significant amount of data for effective training to capture complex PDEs. However, generating high-resolution turbulence simulations is costly, resulting in data shortages that can diminish FNO's performance, particularly in complex 3D scenarios. Secondly, FNO

---

[1] https://drive.google.com/drive/folders/1ldtvSccN8wp9yDl_r1j5RuFmmaBd1CAO?usp=sharing

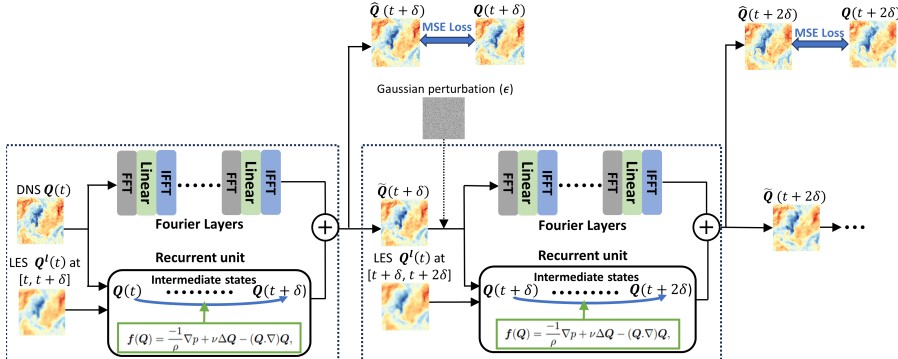

Figure 2: The overall structure of the PENO method with self-augmentation mechanism.

tends to filter out high-frequency information of turbulent flow. This filtration process results in the loss of crucial flow patterns as illustrated in Figure 1. More analysis will be provided in Section 3.2.

## 3 PROPOSED METHOD

In this section, we introduce the proposed PENO method, as outlined in Figure 2. The key component of PENO is a temporal modeling structure that combines FNO and the knowledge from the PDE of turbulent transport. In addition, a new self-augmentation mechanism is designed to ensure that fine-level flow behaviors, especially those present in high-frequency data, are preserved by the Fourier layers in long-term simulation.

### 3.1 PHYSICS-ENHANCED NEURAL OPERATOR

The PENO method sequentially processes input DNS data $\mathbf{Q}(t)$ at each time step and predicts the DNS data for the next step $\hat{\mathbf{Q}}(t+\delta)$. As shown in Figure 2, the prediction at each time step combines the outputs from two network branches, i.e., $\hat{\mathbf{Q}}_{\text{FNO}}(t+\delta)$ and $\hat{\mathbf{Q}}_{\text{PDE}}(t+\delta)$, as $\hat{\mathbf{Q}}(t+\delta) = w_f \hat{\mathbf{Q}}_{\text{FNO}}(t+\delta) + w_p \hat{\mathbf{Q}}_{\text{PDE}}(t+\delta)$, where $w_f$ and $w_p$ are learnable parameters. The first branch consists of several Fourier layers, each of which contains a Fourier transformation, a linear transformation layer, and an inverse Fourier transformation. Although the Fourier layers are based on the Green function method for solving PDEs, they are agnostic of physical knowledge for the target dataset and remain a purely data-driven approach. This leads to the limitation in capturing complex flow dynamics given scarce training data. Hence, we introduce an additional PDE-enhancement branch to complement the simulation by FNO by leveraging underlying PDEs.

Several methods have been developed to incorporate PDEs into the learning process, including the physics-based loss function (Chen et al., 2021; Yousif et al., 2022; Bode et al., 2021; Yousif et al., 2021; Pawar, 2022) and physics-based model structures (Bao et al., 2022; Chen et al., 2021). In this work, the design of the PDE-enhancement network branch is inspired by (Bao et al., 2022), which introduces a new recurrent unit to gradually estimate the temporal gradients over time based on the PDE. The key idea is to leverage the continuous physical relationships depicted by the underlying partial differential equation (PDE) to bridge the gap between discrete data samples and the continuous dynamics of the flow. It also does not require modification of the loss function, which often leads to the training instability for complex PDEs (Sun et al., 2022) and also may not guarantee consistency with PDE in the testing phase (e.g., future simulations).

Specifically, the PDE-enhancement network branch utilizes the Runge–Kutta (RK) discretization method (Butcher, 2007) for PDEs. The PDE for the target variables $\mathbf{Q}$ can be formulated as: $\mathbf{Q}_t = \mathbf{f}(t, \mathbf{Q}; \theta)$, where $\mathbf{Q}_t$ denotes the temporal derivative of $\mathbf{Q}$, and $\mathbf{f}(t, \mathbf{Q}; \theta)$ is a non-linear function determined by the parameter $\theta$. This function summarizes the present value of $\mathbf{Q}$ along with its spatial fluctuations. The turbulent data adheres to the Navier-Stokes equation for an incompressible flow. For example, the dynamics of the velocity field can be expressed by the following PDE:

$$\mathbf{f}(\mathbf{Q}) = -\frac{1}{\rho}\nabla p + \nu \Delta \mathbf{Q} - (\mathbf{Q} \cdot \nabla)\mathbf{Q}, \tag{1}$$

where $\nabla$ signifies the gradient operator, and $\Delta = \nabla \cdot \nabla$ is applied individually to each component of velocity.

Figure 3 illustrates the recurrent unit in the PDE-enhancement network branch, which involves a series of intermediate states $\{\mathbf{Q}(t,0), \mathbf{Q}(t,1), \mathbf{Q}(t,2), \ldots, \mathbf{Q}(t,N)\}$, where $\mathbf{Q}(t,0) \equiv \mathbf{Q}(t)$. The temporal gradients are estimated at these states as $\{\mathbf{Q}_{t,0}, \mathbf{Q}_{t,1}, \mathbf{Q}_{t,2}, \ldots, \mathbf{Q}_{t,N}\}$. Starting from $n = 0$ to $N$, the unit modifies $\mathbf{Q}(t)$ in the gradient's direction $(\mathbf{Q}_{t,n})$ to create the next intermediate state $\mathbf{Q}(t,n)$. We adopt the fourth-order Runge-Kutta method, i.e., $N = 3$. In more detail, we estimate temporal derivatives using the function $\mathbf{f}(\cdot)$. As shown in Eq.(1), to compute $\mathbf{f}(\cdot)$

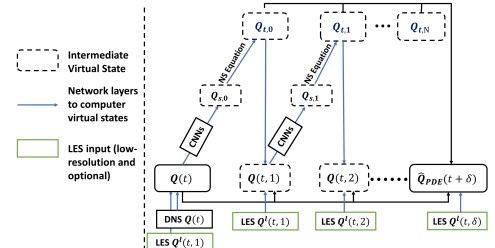

Figure 3: The recurrent unit based on Naiver Stoke equation for reconstructing turbulent flow data in the spatio-temporal field. $\mathbf{Q}_{s,n}$ and $\mathbf{Q}_{t,n}$ represent the spatial and temporal derivatives, respectively, at each intermediate time step.

accurately, it is essential to explicitly estimate both first-order and second-order spatial derivatives. Here we build convolutional network layers to estimate spatial derivatives. After computing the first-order and second-order spatial derivatives, they are incorporated into Eq.(1) to calculate the temporal derivative $\mathbf{Q}_{t,n}$.

Ultimately, we aggregate all intermediate temporal derivatives into a combined gradient for computing the final prediction of the next step's flow data $\hat{\mathbf{Q}}_{\text{PDE}}(t + \delta)$, as $\hat{\mathbf{Q}}_{\text{PDE}}(t + \delta) = \mathbf{Q}(t) + \sum_{n=0}^{N} w_n \mathbf{Q}_{t,n}$, where $\{w_n\}_{n=1}^{N}$ are trainable model parameters.

This model can be further enhanced by leveraging available LES data. At the initial data point $\mathbf{Q}(t)$, we can merge DNS and LES data as $\mathbf{Q}(t) = W^d \mathbf{Q}(t) + W^l \mathbf{Q}^l(t)$, where $W^d$ and $W^l$ are trainable model parameters. Moreover, LES data can often be produced more frequently than DNS data. With the availability of frequent LES data, the intermediate states $\mathbf{Q}(t,n)$ can also enhanced using LES data $\mathbf{Q}^l(t,n)$, formulated as $\mathbf{Q}(t,n) = W^d \mathbf{Q}(t,n) + W^l \mathbf{Q}^l(t,n)$. Following the 4-th order Runga-Kutta method (as detailed in the appendix), LES data $\mathbf{Q}^l(t,n)$ are selected as $\mathbf{Q}^l(t,0) = \mathbf{Q}^l(t)$, $\mathbf{Q}^l(t,1) = \mathbf{Q}^l(t + \delta/2)$, $\mathbf{Q}^l(t,2) = \mathbf{Q}^l(t + \delta/2)$, and $\mathbf{Q}^l(t,3) = \mathbf{Q}^l(t + \delta)$.

## 3.2 Self-augmentation Mechanism

Here we re-examine the frequency spectrum obtained through PENO for modeling turbulent transport. In most PDEs, the large-scale, low-frequency components usually possess larger magnitudes than the small-scale, high-frequency components. Therefore, as regularization, the Fourier layers incorporate a frequency truncation process in each layer, allowing only the lowest $K$ Fourier modes to propagate input information. This truncation process encourages the learning of low-frequency components in PDEs, a phenomenon closely related to the well-known implicit spectral bias (Cao et al., 2019). This bias indicates that neural networks when trained using gradient descent, tend to prioritize learning low-frequency functions (Rahaman et al., 2019). It provides an implicit regularization effect that encourages a neural network to converge to a low-frequency and 'simple' solution.

However, in turbulent flow simulations, it is often observed that high-frequency components carry important flow patterns necessary for accurate prediction. The absence of high-frequency information makes it challenging to adequately represent vital flow data details, leading to inaccurate simulations. As demonstrated in Figure 4 (a) and Figure 4 (c), there is a noticeable difference in the frequency distribution between the FNO's reconstructed data and the target DNS data for the forced isotropic

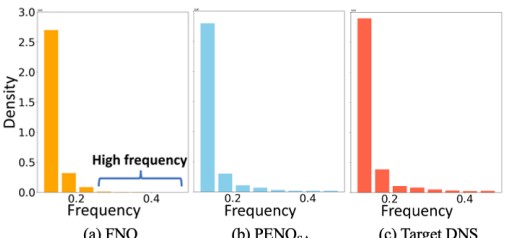

Figure 4: The frequency distribution of the FNO and $\text{PENO}_{\text{SA}}$ predictions, and the target DNS.

flow. It is evident that high-frequency information is missing when the frequency exceeds $0.25$. This absence of high-frequency information results in failures to capture small-scale physical patterns, as illustrated in Figure 1. Therefore, to address the limitation of FNO, a new self-augmentation process is introduced to ensure that vital patterns contained in high-frequency domains can be preserved during the Fourier layer processing.

The self-augmentation process is also shown in Figure 2. Initially, the DNS input $\mathbf{Q}(t)$, is fed to the Fourier layers, yielding an output $\hat{\mathbf{Q}}_{\text{FNO}}(t + \delta)$ at time $t + \delta$ and in the same resolution as the original DNS input $\mathbf{Q}(t)$. Before feeding the output to the next step, we propose to augment it in the high-frequency spectrum so the fine-level flow patterns can be preserved after the Fourier layers in the next step. This augmentation process will leverage a zero-shot upscaling process using the proposed network structure, and do not require auxiliary information.

Specifically, we leverage the capability of FNO in simulating data over different scales in a zero-shot fashion (Li et al., 2020). This can be achieved by altering the output grids in the inverse Fourier transformation. Utilizing the capability of FNO, We create output in a higher resolution, which is represented by $\tilde{\mathbf{Q}}_{\text{FNO}}(t + \delta)$. Concurrently, the PDE-enhancement branch can employ the implicit neural representation method (Chen et al., 2022) to upscale the CNN embeddings, and subsequently generate upscaled outputs $\tilde{\mathbf{Q}}_{\text{PDE}}(t + \delta)$ at the same resolution with $\tilde{\mathbf{Q}}_{\text{FNO}}(t + \delta)$. Finally, we merge the outputs from two branches to create two versions of simulation at $t + \delta$, i.e., $\hat{\mathbf{Q}}(t + \delta)$ in the target resolution, and $\tilde{\mathbf{Q}}(t + \delta)$ at the higher resolution. This process can be summarized as follows:

$$\begin{aligned}
\hat{\mathbf{Q}}(t + \delta) &= w_p \hat{\mathbf{Q}}_{\text{PDE}}(t + \delta) + w_f \hat{\mathbf{Q}}_{\text{FNO}}(t + \delta), \\
\tilde{\mathbf{Q}}(t + \delta) &= \tilde{w}_p \tilde{\mathbf{Q}}_{\text{PDE}_h}(t + \delta) + \tilde{w}_f \tilde{\mathbf{Q}}_{\text{FNO}}(t + \delta),
\end{aligned} \tag{2}$$

where $w_p$, $\tilde{w}_p$, $w_f$, and $\tilde{w}_f$ are trainable model parameters.

This sequential prediction process will be repeated throughout the entire simulation period. The obtained sequence $\{\hat{\mathbf{Q}}(t)\}$ in the original resolution are the final simulation outputs. The training loss will be defined on this sequence during the training period, i.e., $\{\hat{\mathbf{Q}}(t)\}_{t=t_0}^{t_0+K\delta}$, based the mean-squared errors, as $\mathcal{L} = \sum_{t \in \mathcal{D}} ||\hat{\mathbf{Q}}(t) - \mathbf{Q}(t)||^2 / |\mathcal{D}|$, where $\mathcal{D}$ is the set of prediction steps in the training set. In our implementation, we create overlapping sequence batches in the training phase.

The upscaled output $\tilde{\mathbf{Q}}$ will serve as an augmented input to the next time step, which helps better preserve the high-frequency information during long-term auto-regressive simulation. Note that the upscaled simulations are generated completely in a zero-shot manner using no additional parameters. Hence, the training loss $\mathcal{L}$ will also help refine $\tilde{\mathbf{Q}}$. To further improve model robustness and mitigate overfitting, we also introduce random Gaussian perturbations, denoted as $\epsilon \sim \mathcal{N}(0, 0.02)$, and incorporate it into $\tilde{\mathbf{Q}}(t + \delta)$ independently for each position $\mathbf{x}$, as: $\tilde{\mathbf{Q}}(\mathbf{x}, t + \delta) = \tilde{\mathbf{Q}}(\mathbf{x}, t + \delta) + \epsilon$.

Then the perturbed upscaled output $\tilde{\mathbf{Q}}(t + \delta)$ is fed into the PENO for the prediction of the next time step $(t + 2\delta)$. This self-augmentation process is repeated for the following time steps. Starting from the second step in a sequence, the Fourier layers and the PDE-enhancement layers will take the perturbed upscaled input and produce the two versions of output simulations $\tilde{\mathbf{Q}}(t)$ and $\hat{\mathbf{Q}}(t)$. Note that here the output $\tilde{\mathbf{Q}}(t)$ is in the same resolution as the input ($\tilde{\mathbf{Q}}(t - \delta)$) while the other output $\hat{\mathbf{Q}}(t)$ is at a lower resolution than the input. The transformation through Fourier layers is agnostic of input and output scales and thus requires no structural changes. For the PDE-enhancement layer, we will utilize the same implicit neural representation method by down-sampling the CNN embeddings.

As illustrated in Figure 4 (b), the high-frequency information is retained by using the proposed method PENO$_{\text{SA}}$ (PENO + self-augmentation mechanism), in contrast to the frequency spectrum of the FNO shown in Figure 4 (a). The proposed method can help fully leverage the power of Fourier layers in selectively filtering over the augmented signals, which can contain a mixture of vital flow patterns and noise factors. Further improvement can be made by introducing Gaussian perturbations that are deliberately designed to improve the model's robustness and generalizability, which we will keep as future work.

## 4 EXPERIMENT

### 4.1 EXPERIMENTAL SETTINGS.

**Datasets.** To assess the effectiveness of the proposed PENO method, we consider two groups of tests. The first group of tests aims to evaluate the simulation performance on each specific 3D flow dataset. We consider two different turbulent flow datasets, the forced isotropic turbulent flow (FIT) (Minping et al., 2012) and the Taylor-Green vortex (TGV) flow (Brachet et al., 1984). In both

Table 1: Quantitative performance (measured by SSIM, and Dissipation difference) on $(u, v, w)$ channels by different methods in the FIT dataset. The performance is measured by the average results of the first 10 time steps.

| Method | SSIM ↑ | Dissipation diff ↓ |
|---|---|---|
| RCAN | (0.881, 0.871, 0.874) | (0.224, 0.225, 0.225) |
| HDRN | (0.887, 0.875, 0.875) | (0.217, 0.223, 0.223) |
| FSR | (0.887, 0.877, 0.875) | (0.218, 0.221, 0.223) |
| DCS/MS | (0.888, 0.878, 0.880) | (0.216, 0.220, 0.214) |
| SRGAN | (0.891, 0.881, 0.215) | (0.215, 0.217, 0.215) |
| CTN | (0.901, 0.891, 0.903) | (0.161, 0.173, 0.174) |
| FNO | (0.912, 0.915, 0.911) | (0.153, 0.151, 0.150) |
| PRU | (0.926, 0.920, 0.926) | (0.145, 0.144, 0.144) |
| PENO | (0.936, 0.935, 0.937) | (0.135, 0.134, 0.136) |
| PENO$_{SR}$ | (0.964, 0.966, 0.965) | (0.120, 0.118, 0.118) |
| **PENO$_{SA}$** | (**0.968**, **0.972**, **0.967**) | (**0.110**, **0.107**, **0.110**) |

cases, the mean velocity is zero, denoted as $\overline{\mathbf{Q}}(t) = 0$, and the Reynolds number is high enough to generate turbulent conditions.

In particular, the FIT dataset contains original DNS records of forced isotropic turbulence, which is an incompressible flow. This flow undergoes energy injection at lower wave numbers as a part of its forcing mechanism. The DNS dataset encompasses $5,024$ time steps, each spaced at intervals of $0.002s$, and includes both velocity and pressure field data. For this study, the DNS data has three distinct grid sizes: $128 \times 64 \times 64$, $128 \times 128 \times 128$, and $128 \times 256 \times 256$. Concurrently, the LES data are produced on $128 \times 32 \times 32$ grids. Both datasets are gathered across 128 uniformly spaced grid points along the $z$ axis.

The Taylor-Green vortex (TGV) represents a different incompressible flow. The evolution of the TGV involves the elongation of vorticity, resulting in the generation of small-scale, dissipating eddies. A box flow scenario is examined within a cubic periodic domain spanning $[-\pi, \pi]$ in all three directions. The DNS and LES resolutions are $128 \times 128 \times 65$ and $32 \times 32 \times 65$. Both of them are produced along the 65 equally-spaced grid points along the $z$ axis.

The second group of tests aims to validate the transferability of the PENO method, and it uses a dataset comprising 100 groups of 2D vorticity simulations (Li et al., 2020) under different viscosity coefficients ranging from $\{1e^{-5}, 1.5e^{-5}\}$. Each group contains a complete sequence of 50 time steps with a time interval of $0.03s$. The DNS and LES resolutions are $128 \times 128$ and $64 \times 64$, respectively. More details of the datasets are described in the appendix.

**PENO and baselines.** The performance of the PENO method is evaluated and compared with multiple existing methods for simulating turbulent transport, including SR-based reconstruction methods and sequential prediction methods. Specifically, the complete PENO$_{SA}$ method (PENO+self-augmentation mechanism) is compared against three popular SR methods RCAN (Zhang et al., 2018a), HDRN (Van Duong et al., 2021), and SRGAN (Ledig et al., 2017), two popular dynamic fluid downscaling methods DCS/MS (Fukami et al., 2019) and FSR (Yang et al., 2023), and sequential prediction methods including a convolutional transition network (CTN) (Bao et al., 2022) created by combining SRCNN (Dong et al., 2014) and LSTM (Hochreiter & Schmidhuber, 1997), and the standard FNO (Li et al., 2020) and PRU (Bao et al., 2022) methods. Comparison against FNO and PRU can help verify the effectiveness of each component in the proposed model.

Besides the complete version PENO$_{SA}$, we also implement two variants of the proposed methods, PENO and PENO$_{SR}$. PENO is developed by directly combining FNO and PDE-enhancement branches but without the self-augmentation mechanism. PENO$_{SR}$ includes the upscaling step in the self-augmentation mechanism but without the addition of Gaussian perturbation. The objective of comparison amongst PENO-based methods is to demonstrate the advantages of the self-augmentation mechanism.

**Experimental designs.** Both the FIT and TGV datasets are utilized to evaluate the effectiveness of PENO-based methods and the baselines in the first group of tests. For the FIT dataset, the models are trained on data spanning a continuous one-second interval with a time step of $\delta = 0.02s$, encompassing a total of 50 time steps. The performance of these trained models is then tested on the subsequent 0.4 second period, which corresponds to 20 time steps. For the TGV dataset, the training process is conducted on a continuous 40-second period, with each time step being $\delta = 2s$, and the subsequent 40 seconds of data are used for evaluation.

Table 2: Quantitative performance (measured by SSIM, and Dissipation difference) on $(u, v, w)$ channels by different methods in the TGV dataset. The performance is measured by the average results of the first 10 time steps.

| Method | SSIM ↑ | Dissipation diff×10 ↓ |
|---|---|---|
| RCAN | (0.627, 0.622, 0.631) | (0.073, 0.074, 0.071) |
| HDRN | (0.638, 0.638, 0.641) | (0.072, 0.072, 0.068) |
| FSR | (0.646, 0.648, 0.649) | (0.070, 0.073, 0.066) |
| DSC/MS | (0.647, 0.649, 0.649) | (0.070, 0.071, 0.065) |
| SRGAN | (0.661, 0.658, 0.666) | (0.068, 0.067,0.058) |
| CTN | (0.623, 0.624, 0.627) | (0.093, 0.096, 0.087) |
| FNO | (0.645, 0.646, 0.648) | (0.072, 0.071, 0.072) |
| PRU | (0.708, 0.705, 0.702) | (0.048, 0.046, 0.043) |
| PENO | (0.721, 0.720, 0.715) | (0.043, 0.044, 0.042) |
| PENO$_{SR}$ | (0.822, 0.825, 0.821) | (0.035, 0.037, 0.036) |
| **PENO$_{SA}$** | (**0.843**, **0.847**, **0.844**) | (**0.032**, **0.033**, **0.034**) |

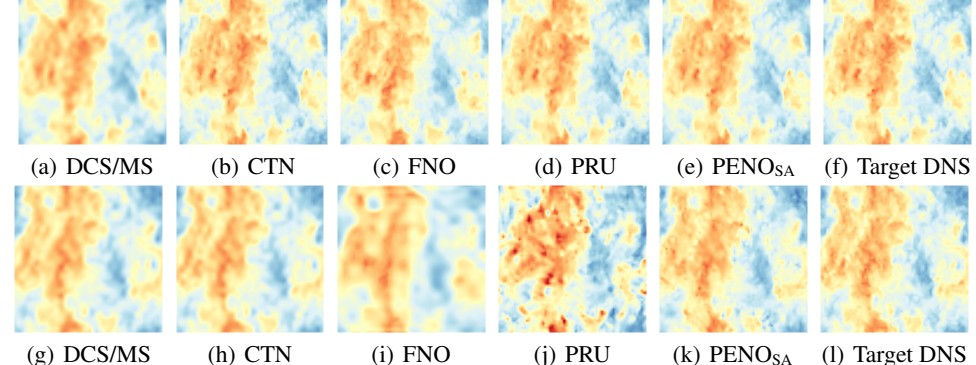

(a) $u$ Channel.  (b) $v$ Channel.  (c) $w$ Channel.

Figure 5: Change of dissipation difference by different models from 1st (5.6s) to 20th (6s) time step in the FIT dataset.

(a) DCS/MS  (b) CTN  (c) FNO  (d) PRU  (e) PENO$_{SA}$  (f) Target DNS

(g) DCS/MS  (h) CTN  (i) FNO  (j) PRU  (k) PENO$_{SA}$  (l) Target DNS

Figure 6: Reconstructed $u$ channel by each method on a sample testing slice along the $z$ dimension in the FIT dataset. The visual results are shown at 1st (5.6s), and 20th (6s) in (a)-(f), and (g)-(l).

To evaluate the transferability of PENO$_{SA}$, a second group of tests using 2D vorticity data is conducted. These tests are divided into three categories: few-shot, zero-shot, and sequential tests. Specifically, few-shot and zero-shot tests aim to testify the model generalizability across turbulent flows governed by different PDEs. They both utilize 50 complete flow sequences (15 seconds over 50 time steps) for training with the viscosity value (a parameter in the Navier-Stokes equation) sampled uniformly from $1e^{-5}$ to $1.25e^{-5}$, followed by testing on 10 additional sequences with the viscosity value sampled uniformly from $1.25e^{-5}$ to $1.5e^{-5}$. The zero-shot test directly applies the sequential prediction models obtained from training sequences to predict vorticity for 20 time steps on each testing sequence. In contrast, the few-shot test also utilizes the first 10 time steps of data from each testing sequence to fine-tune the trained model before proceeding with sequential predictions in the next 20 time steps. Different from both zero-shot and few-shot tests, the sequential test aims to testify the model generalizability over time. It trains the models using the first 20 time steps from 50 complete training sequences and then applies the obtained models to create simulations for the following 20 time steps in the same set of flow sequences.

The assessment of DNS simulation performance employs two metrics: the structural similarity index measure (SSIM) (Wang et al., 2004) and dissipation difference (Wikipedia contributors, 2022). SSIM measures the similarity between the reconstructed and target DNS data in terms of luminance, contrast, and overall structure. Higher SSIM values indicate better performance. Dissi-

pation evaluates the model's gradient capturing ability, considering dissipation for each velocity vector component ($u$, $v$, and $w$). The dissipation operator is defined by $\chi(Q) \equiv \nabla Q \cdot \nabla Q = \left(\frac{\partial Q}{\partial x}\right)^2 + \left(\frac{\partial Q}{\partial y}\right)^2 + \left(\frac{\partial Q}{\partial z}\right)^2$. The dissipation is used to measure the difference in flow gradient between the true DNS and generated data. This is represented by $|\chi(\mathbf{Q}) - \chi(\hat{\mathbf{Q}})|$, and the smaller difference indicates better performance in capturing spatial variations in turbulence. More details of the experimental settings are described in the appendix.

### 4.2 Performance on a Single 3D Flow Dataset

**Quantitative results.** Tables 1 and 2 summarize the average performance over the first 10 time steps during the testing phase, evaluated on both the FIT and TGV datasets. Compared to baseline methods, PENO-based methods consistently show superior performance on both datasets, with the highest SSIM values and the lowest dissipation differences. Several highlights also emerge: (1) SR-based baselines such as RCAN, DCS/MS, and FSR, have inferior performance in terms of SSIM and dissipation differences, which indicates that they are unable to recover fine-level flow patterns in DNS. (2) The comparison between FNO and PENO highlights the improvement achieved by integrating the physical knowledge of PDEs into FNO's learning process. (3) PRU generally performs better than FNO for modeling complex turbulence due to the awareness of underlying physics. PRU performs worse than the proposed PENO method because it can easily create artifacts over long-term simulation, which we will discuss later in other results. (4) The comparison between PENO, PENO$_{SR}$ and PENO$_{SA}$ reveals improvement through the incorporation of a self-augmentation mechanism, especially for both the upscaling step and the addition of random Gaussian perturbations.

**Temporal analysis.** In Figure 6, we evaluate the performance for simulating DNS at each step over a 0.4s period (20 time steps) during the testing phase on the FIT dataset. We measure the performance change using dissipation difference, as presented in Figure 5. Several observations are highlighted: (1) As the gap between the training period and the testing time step increases, there is a general decline in model performance for all the methods. It can be seen that PENO$_{SA}$ has a relatively stable performance in long-term prediction, outperforming other methods in terms of accuracy. (2) The comparison amongst FNO, PRU, and PENO$_{SA}$ indicates that the integration of physical knowledge and the use of self-augmentation mechanisms in PENO$_{SA}$ effectively capture turbulence dynamics, which helps reduce accumulated errors in long-term simulations. (3) FNO struggles to achieve good performance starting from early testing phase. Although it achieves lower errors than many other methods, we will show that it actually oversmooths the simulation and fails to capture fine-level patterns. More details of the temporal analysis are also described in the appendix.

**Validation via physical metrics.** We also assess the temporal simulations based on their turbulent kinetic energy, which is a critical property for verifying the accuracy of the simulations. Figure 7 displays the energy levels associated with the target DNS, as well as the flow data simulated by both the baseline models and PENO$_{SA}$ within the FIT dataset. Notable observations include: (1) PENO$_{SA}$ shows improved performance compared to the baseline models, closely mirroring target DNS's energy transport accurately. (2) FNO struggles to adhere to the correct energy transport trend after the 5th time step. PRU also achieves good performance in preserving the energy due to the awareness of physics. Meanwhile, DCS/MS and CTN largely fail to accurately capture the energy transport pattern from the 1st test point.

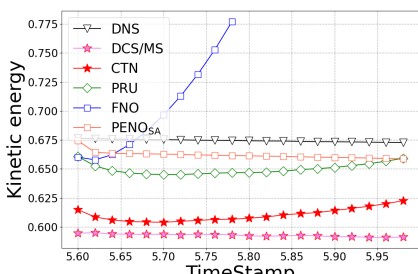

Figure 7: Change of kinetic energy produced by the real DNS and different models in the FIT dataset.

**Visualization.** the simulated flow data for the FIT dataset are displayed at multiple time steps (1st and 20th) following the training period. For each time step, slices of the $w$ component at a specified $z$ value are presented. Several conclusions are highlighted: (1) At the 1st step, PENO$_{SA}$, PRU, FNO, and CTN obtain good performance because the test data closely resemble the training data at the last time step. In contrast, the baseline DSC/MS leads to poor performance starting from early time. (2) At the 20th time step after the training phase, PENO$_{SA}$ significantly outperform FNO and PRU. Specifically, FNO is unable to capture fine-level flow patterns due to the loss of high-frequency signals. While PRU is capable of capturing the complex transport patterns but introduces structural distortions and random artifacts due to accumulated errors in long-term simulations. In contrast,

PENO$_{\text{SA}}$ addresses these issues effectively, resulting in significantly improved performance in long-term simulation. More details of visual results are described in the appendix.

**Performance in simulating at different resolutions.** Similar to FNO, PENO$_{\text{SA}}$ can also create simulations at a resolution different from that of the training data. We evaluate the performance of PENO$_{\text{SA}}$ and FNO on the FIT dataset, training all models at a resolution of $128 \times 64 \times 64$ and testing them at varying resolutions: $128 \times 64 \times 64$, $128 \times 128 \times 128$, and $128 \times 256 \times 256$. Both methods employ zero-shot super-resolution techniques (Shocher et al., 2018) without using DNS data at the target higher resolution for tuning. Figure 8 (a) and (b) show the comparative performance of both methods. Both PENO$_{\text{SA}}$ and FNO faces increased challenges in accurately reproducing flow data at higher resolutions, attributed to the augmented complexity present in finer-scale flow patterns. However, PENO$_{\text{SA}}$ can achieve better performance in zero-shot super-resolution in terms of SSIM and dissipation difference metrics. In contrast, FNO performs worse in rendering precise predictions at equivalent resolutions and also in generalizing to unseen resolutions. These findings underscore PENO$_{\text{SA}}$'s superiority in creating simulations over long periods and at different resolutions.

In addition, the ablation study for utilizing LES data is also conducted, the experimental analysis is shown in appendix.

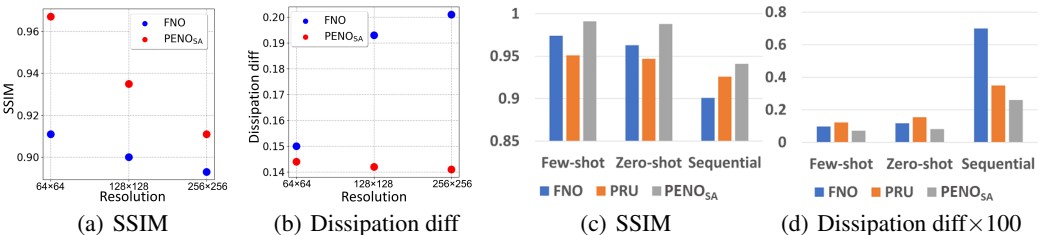

|  (a) SSIM  |  (b) Dissipation diff  |  (c) SSIM  |  (d) Dissipation diff $\times 100$ |

Figure 8: (a) and (b) show the quantitative performance of the models in the $w$ channel, evaluated across different resolutions in the FIT data. (c) and (d) show the average performance over the first 20 time steps in the 2D vorticity data, which is used for validating the models' transferability.

### 4.3 TRANSFERABILITY

To assess the transferability of PENO$_{\text{SA}}$, we evaluate the performance of PENO$_{\text{SA}}$, FNO, and PRU on the 2D vorticity dataset. Figure 8 (c) and (d) illustrate the performance of these models in three tests: few-shot test, zero-shot test, and sequential test. From this comparison, two conclusions are drawn. Firstly, PENO$_{\text{SA}}$ surpasses both FNO and PRU in all tests. It demonstrates that PENO$_{\text{SA}}$ can generalize not only to different time periods but also to different PDE-governed flow sequences. It also achieves good performance under both few-shot and zero-shot scenarios. Secondly, FNO surpasses PRU in both few-shot and zero-shot tests, as FNO better captures generalizable flow patterns from long sequences of training data. These learned patterns can be better transferred to other testing sequences. However, PRU outperforms FNO in the sequential test. This is because FNO cannot easily capture flow patterns from only a small portion of the training data sequences. In contrast, PRU can utilize the known PDE format to more accurately capture complete flow patterns and achieve better predictive performance. We also present the visual results in the appendix to indicate the superiority of PENO$_{\text{SA}}$.

### 5 CONCLUSION

A novel physics-enhanced neural operator (PENO) has been developed to improve the simulation of turbulent transport over long-term simulations and various flow datasets. PENO is particularly applicable in the domain of unsteady, incompressible, Newtonian turbulent flows under conditions of spatial homogeneity. Specifically, PENO integrates physical knowledge of PDEs with the FNO framework to effectively model turbulence dynamics. Additionally, PENO introduces a novel self-augmentation mechanism designed to reduce the accumulation of errors in long-term simulations. The efficacy of the model is assessed through three turbulent flow configurations, employing both flow visualization and statistical analysis techniques. The experimental results confirm PENO's enhanced capabilities in long-term simulations. More significantly, the PENO method shows potential for broad applicability in scientific problems characterized by complex temporal dynamics, particularly where generating high-resolution simulations is prohibitively expensive.

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

## A  RUNGE-KUTTA METHOD FOR PDE ENHANCEMENT

The principal idea of the Runge-Kutta (RK) discretization method (Butcher, 2007) is to use the continuous relationships outlined by the underlying PDEs to connect discrete data points with the continuous flow dynamics. This approach is adaptable to any dynamic system that is defined by deterministic PDEs. The PDE that describes the target variables $\mathbf{Q}$ as expressed by:

$$\mathbf{Q}_t = \mathbf{f}(t, \mathbf{Q}; \theta), \tag{3}$$

where $\mathbf{Q}_t$ denotes the temporal derivative of $\mathbf{Q}$, and $\mathbf{f}(t, \mathbf{Q}; \theta)$ is a non-linear function determined by the parameter $\theta$. This function summarizes the present value of $\mathbf{Q}$ along with its spatial fluctuations. The turbulent data adheres to the Navier-Stokes equation for an incompressible flow. For example, the dynamics of the velocity field can be expressed by the following PDE:

$$\mathbf{f}(\mathbf{Q}) = -\frac{1}{\rho}\nabla p + \nu\Delta\mathbf{Q} - (\mathbf{Q} \cdot \nabla)\mathbf{Q}, \tag{4}$$

where the term $\nabla$ represents the gradient operator, and $\Delta = \nabla \cdot \nabla$ acts on each component of the velocity vector. We omit the independent variable $t$ in the function $\mathbf{f}(\cdot)$ because $\mathbf{f}(\mathbf{Q})$ in the Navier-Stokes equations refers to a specific time $t$, analogous to the $t$ in $\mathbf{Q}_t$. Figure 9 illustrates the overall structure, which involves a series of intermediate states $\{\mathbf{Q}(t, 0), \mathbf{Q}(t, 1), \mathbf{Q}(t, 2), \ldots, \mathbf{Q}(t, N)\}$, where $\mathbf{Q}(t, 0) \equiv \mathbf{Q}(t)$. The temporal gradients are estimated at these states as $\{\mathbf{Q}_{t,0}, \mathbf{Q}_{t,1}, \mathbf{Q}_{t,2}, \ldots, \mathbf{Q}_{t,N}\}$. Beginning with $\mathbf{Q}(t, 0) = \mathbf{Q}(t)$, we estimate the temporal gradient $\mathbf{Q}_{t,0}$, then progresses $\mathbf{Q}(t)$ in the direction of this gradient to generate the subsequent intermediate state $\mathbf{Q}(t, 1)$. This procedure is iterated for $N$ intermediate states. For the fourth-order RK method, which is applied here, we have $N = 3$.

To initiate with the data point $\mathbf{Q}(t)$, we employ an augmentation by integrating LES with DNS data, formulated as $\mathbf{Q}(t) = W^d\mathbf{Q}(t) + W^l\mathbf{Q}^l(t)$, where $W^d$ and $W^l$ are trainable model parameters, and $\mathbf{Q}^l(t)$ is the up-sampled LES data with the same resolution as DNS. We estimate the first temporal gradient $\mathbf{Q}_{t,0} = \mathbf{f}(\mathbf{Q}(t))$ using the Navier-Stokes equation and computes the next intermediate state variable $\mathbf{Q}(t, 1)$ by moving the flow data $\mathbf{Q}(t)$ along the direction of temporal derivatives. Given frequent LES data, the intermediate states $\mathbf{Q}(t, n)$ are also augmented by using LES data $\mathbf{Q}^l(t, n)$, as $\mathbf{Q}(t, n) = W^d\mathbf{Q}(t, n) + W^l\mathbf{Q}^l(t, n)$. This iterative method progresses $\mathbf{Q}(t)$ along the computed gradient $\mathbf{Q}_{t,n}$ to compute the next intermediate states $\mathbf{Q}(t, n + 1)$, expressed as:

$$\begin{aligned}
\mathbf{Q}(t, 1) &= \mathbf{Q}(t) + \delta\frac{\mathbf{Q}_{t,0}}{2}, \\
\mathbf{Q}(t, 2) &= \mathbf{Q}(t) + \delta\frac{\mathbf{Q}_{t,1}}{2}, \\
\mathbf{Q}(t, 3) &= \mathbf{Q}(t) + \delta\mathbf{Q}_{t,2}.
\end{aligned} \tag{5}$$

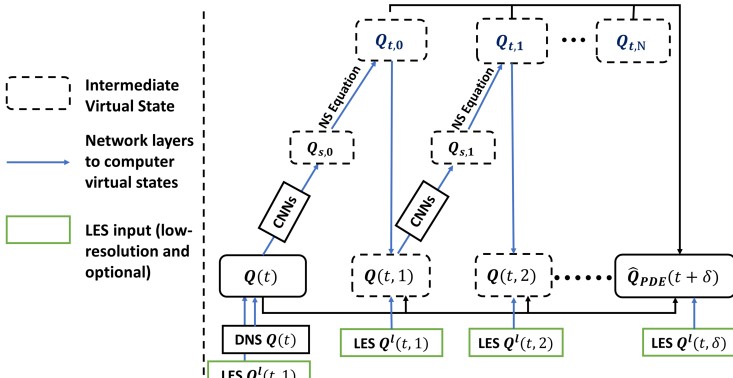

Figure 9: The recurrent unit based on Naiver Stoke equation for reconstructing turbulent flow data in the spatio-temporal field. $\mathbf{Q}_{s,n}$ and $\mathbf{Q}_{t,n}$ represent the spatial and temporal derivatives, respectively, at each intermediate time step.

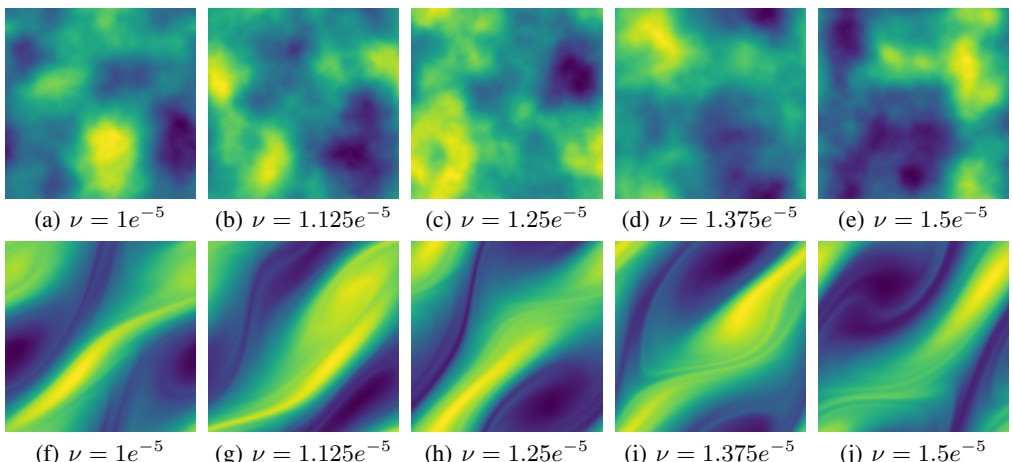

(a) $\nu = 1e^{-5}$    (b) $\nu = 1.125e^{-5}$    (c) $\nu = 1.25e^{-5}$    (d) $\nu = 1.375e^{-5}$    (e) $\nu = 1.5e^{-5}$

(f) $\nu = 1e^{-5}$    (g) $\nu = 1.125e^{-5}$    (h) $\nu = 1.25e^{-5}$    (i) $\nu = 1.375e^{-5}$    (j) $\nu = 1.5e^{-5}$

Figure 10: 2D vorticity samples from different groups of DNS flow sequences with varying viscosities $\nu$. Samples (a)-(e) are from the initial stages, and samples (f)-(j) are from the final stages.

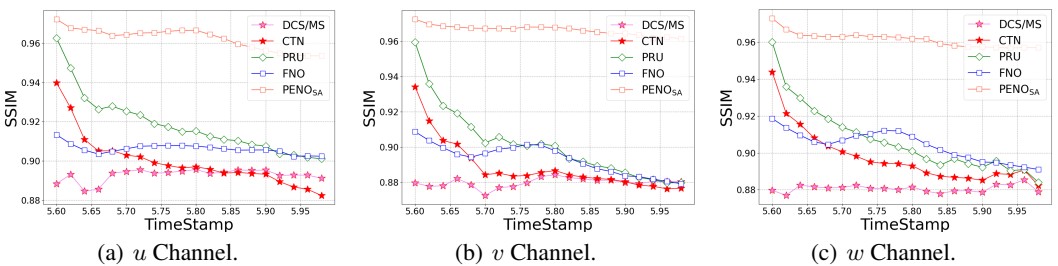

(a) $u$ Channel.      (b) $v$ Channel.      (c) $w$ Channel.

Figure 11: Change of SSIM value by different models from 1st (5.6s) to 20th (6s) time step in the FIT dataset.

The temporal gradient at the final intermediate stage, $\mathbf{Q}_{t,3}$, is derived using $\mathbf{f}(\mathbf{Q}(t,3))$. Referring to Eq,(5), selections for intermediate LES data, $\mathbf{Q}^l(t,n)$, are specified as follows: $\mathbf{Q}^l(t,1)$ and $\mathbf{Q}^l(t,2)$ are set to $\mathbf{Q}^l(t+\delta/2)$, while $\mathbf{Q}^l(t,3)$ corresponds to $\mathbf{Q}^l(t+\delta)$. Ultimately, we aggregate all intermediate temporal derivatives into a combined gradient for computing the final prediction of

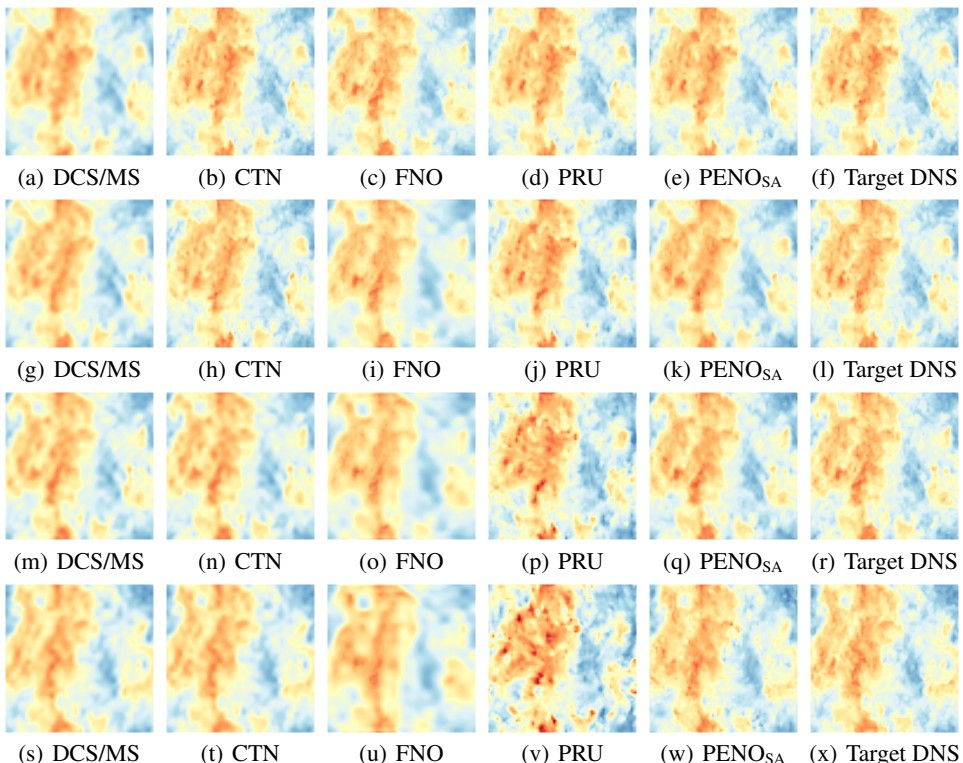

(a) DCS/MS    (b) CTN    (c) FNO    (d) PRU    (e) PENO$_{\text{SA}}$    (f) Target DNS

(g) DCS/MS    (h) CTN    (i) FNO    (j) PRU    (k) PENO$_{\text{SA}}$    (l) Target DNS

(m) DCS/MS    (n) CTN    (o) FNO    (p) PRU    (q) PENO$_{\text{SA}}$    (r) Target DNS

(s) DCS/MS    (t) CTN    (u) FNO    (v) PRU    (w) PENO$_{\text{SA}}$    (x) Target DNS

Figure 12: Reconstructed $u$ channel by each method on a sample testing slice along the $z$ dimension in the FIT dataset. The visual results are shown at 1st (5.6s), 5th (5.7s), 10th (5.8s) and 20th (6s) in (a)-(f), (g)-(l), (m)-(r) and (s)-(x), respectively.

the next step's flow data $\hat{\mathbf{Q}}_{\text{PDE}}(t + \delta)$, as:

$$\hat{\mathbf{Q}}_{\text{PDE}}(t + \delta) = \mathbf{Q}(t) + \sum_{n=0}^{N} w_n \mathbf{Q}_{t,n}. \tag{6}$$

where $\{w_n\}_{n=1}^{N}$ are trainable model parameters.

In more detail, the model estimates temporal derivatives using the function $\mathbf{f}(\cdot)$. As shown in Eq.(4), to compute $\mathbf{f}(\cdot)$ accurately, it's essential to explicitly estimate both first-order and second-order spatial derivatives. This estimation of spatial derivatives is executed by convolutional neural network layers (CNNs) (Bao et al., 2022). After computing the first-order and second-order spatial derivatives, they are incorporated into Eq.(4) to calculate the temporal derivative $\mathbf{Q}_{t,n}$.

## B   EXPERIMENT

### B.1   DATASET

To assess the effectiveness of the proposed PENO method, we consider two groups of tests. The first group of tests aims to evaluate the simulation performance on each specific 3D flow dataset. We consider two different turbulent flow datasets, the forced isotropic turbulent flow (FIT) (Minping et al., 2012) and the Taylor-Green vortex (TGV) flow (Brachet et al., 1984). In both cases, the mean velocity is zero, denoted as $\overline{\mathbf{Q}}(t) = 0$, and the Reynolds number is high enough to generate turbulent conditions.

The FIT dataset comprises the original DNS records of forced isotropic turbulence, representing an incompressible flow.The flow is subjected to energy injection at low wave numbers as part of the forcing mechanism. The DNS data consists of $5024$ time steps, with each step separated by a time

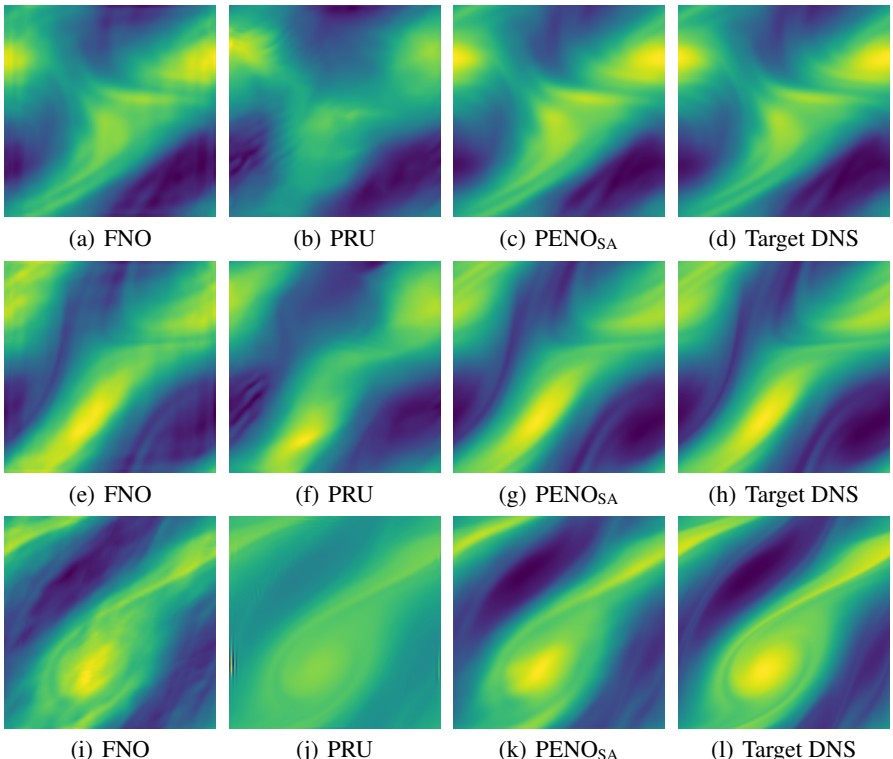

(a) FNO     (b) PRU     (c) PENO$_{SA}$     (d) Target DNS

(e) FNO     (f) PRU     (g) PENO$_{SA}$     (h) Target DNS

(i) FNO     (j) PRU     (k) PENO$_{SA}$     (l) Target DNS

Figure 13: Reconstructed 2D flow in the vorticity field by each method. The visual results are shown at the 20th time step of the testing phase from the sequential test. (a)-(d), (e)-(h), and (i)-(l) correspond to three different groups of results, respectively.

Table 3: The performance of FNO and PENO$_{SA}$ on FIT dataset with and without using LES input.

| Method | LES input | SSIM ↑ | Dissipation diff ↓ |
|---|---|---|---|
| FNO | NO | (0.912, 0.915, 0.911) | (0.153, 0.151, 0.150) |
| **FNO** | **YES** | **(0.923, 0.925, 0.924)** | **(0.144, 0.142, 0.141)** |
| PENO$_{SA}$ | NO | (0.954, 0.953, 0.954) | (0.122, 0.124, 0.123) |
| **PENO$_{SA}$** | **YES** | **(0.968, 0.972, 0.967)** | **(0.110, 0.107, 0.110)** |

Table 4: PENO$_{SA}$'s performance (measured by SSIM, and Dissipation difference) of on $(u, v, w)$ channels by different levels of random Gaussian noise $\mathcal{N}$ in the FIT dataset. The performance is measured by the average results of the first 10 time steps.

| $\mathcal{N}$ | SSIM ↑ | Dissipation diff ↓ |
|---|---|---|
| $\mathcal{N}(0, 0.01)$ | (0.964, 0.966, 0.965) | (0.120, 0.118, 0.118) |
| $\mathcal{N}(0, 0.02)$ | (0.968, 0.972, 0.967) | (0.110, 0.107, 0.110) |
| $\mathcal{N}(0, 0.05)$ | **(0.971, 0.972, 0.970)** | **(0.108, 0.107, 0.108)** |
| $\mathcal{N}(0, 0.10)$ | (0.974, 0.974, 0.974) | (0.106, 0.105, 0.106) |
| $\mathcal{N}(0, 0.15)$ | (0.971, 0.970, 0.971) | (0.109, 0.110, 0.109) |
| $\mathcal{N}(0, 0.20)$ | (0.965, 0.965, 0.966) | (0.117, 0.116, 0.117) |

interval of 0.002s, encompassing both velocity and pressure fields. For this study,the DNS data has three different grids: $128 \times 64 \times 64$, $128 \times 128 \times 128$, and $128 \times 256 \times 256$. Simultaneously, the LES data is generated on grids of size $128 \times 32 \times 32$. Both DNS and LES data are collected along the 128 equally spaced grid points along the $z$ axis.

The Taylor-Green vortex (TGV) represents another incompressible flow. The evolution of the TGV involves the elongation of vorticity, resulting in the generation of small-scale, dissipating eddies. A box flow scenario is examined within a cubic periodic domain spanning $[-\pi, \pi]$ in all three direc-

Table 5: PENO$_{SA}$'s performance (measured by SSIM, and Dissipation difference) of on $(u, v, w)$ channels by different levels of random Gaussian noise $\mathcal{N}$ in the TGV dataset. The performance is measured by the average results of the first 10 time steps.

| $\mathcal{N}$ | SSIM $\uparrow$ | Dissipation diff $\times 10 \downarrow$ |
|---|---|---|
| $\mathcal{N}(0, 0.01)$ | (0.824, 0.826, 0.824) | (0.034, 0.036, 0.035) |
| $\mathcal{N}(0, 0.02)$ | (0.843, 0.847, 0.844) | (0.032, 0.033, 0.034) |
| $\mathcal{N}(0, 0.05)$ | (0.847, 0.849, 0.846) | (0.031, 0.031, 0.032) |
| $\mathcal{N}(0, 0.10)$ | (0.851, 0.852, 0.853) | (0.030, 0.030, 0.029) |
| $\mathcal{N}(0, 0.15)$ | **(0.852, 0.853, 0.852)** | **(0.029, 0.029, 0.030)** |
| $\mathcal{N}(0, 0.20)$ | (0.839, 0.842, 0.839) | (0.034, 0.033, 0.034) |

tions. The initial conditions are defined as:

$$
\begin{aligned}
u(x, y, z, 0) &= \sin(x) \cos(y) \cos(z), \\
v(x, y, z, 0) &= -\cos(x) \sin(y) \cos(z), \\
w(x, y, z, 0) &= 0.
\end{aligned}
\tag{7}
$$

The DNS and LES resolutions are $128 \times 128 \times 65$ and $32 \times 32 \times 65$, respectively. Both DNS and LES data are produced along the 65 equally-spaced grid points along the $z$ axis.

The second group of tests aims to validate the transferability of the PENO method. Here we examine the 2D Navier-Stokes equation in vorticity form (Li et al., 2020), which applies to a viscous and incompressible fluid, described as:

$$
\begin{aligned}
\partial_t w(x, t) + u(x, t) \cdot \nabla w(x, t) &= \nu \Delta w(x, t) + f(x) \\
\nabla \cdot u(x, t) &= 0 \\
w(x, 0) &= w_0(x)
\end{aligned}
\tag{8}
$$

where $u$ represents the velocity field. The vorticity, denoted by $w$, is defined as the curl of the velocity field, $w = \nabla \times u$. The initial vorticity is given by $w_0$. Additionally, $\nu$ is the viscosity coefficient, and $f$ represents the forcing function. For the simulation, 100 groups of vorticity flow data sequences are used, each under different initial conditions and with viscosity coefficients $\nu$ ranging from $\{1e^{-5}, 1.5e^{-5}\}$ are used. Each group consists of a complete sequence of 50 time steps, with a time interval of $0.03s$. The DNS and LES resolutions are $128 \times 128$ and $64 \times 64$, respectively. Figure 10 displays various samples from different groups of DNS flow sequences with varying viscosities $\nu$.

## B.2 Implementation Details

Data normalization is conducted on both the training and testing datasets to normalize to the range [0,1]. Then, PENO is implemented using PyTorch 2.12 on an A100 GPU. The model undergoes training for 500 epochs with the ADAM optimizer (Kingma & Ba, 2014). The initial learning rate is set at 0.001. All hidden variables are in 16 dimensions. In the FNO branch, the number of Fourier layers is established at 3, while in the PDE-enhancement branch, the number of CNN layers is fixed at 2 for calculating spatial derivatives.

## B.3 Performance on a Single 3D Flow Dataset

**Temporal analysis.** We evaluate the performance for simulating DNS at each step over a 0.4s period (20 time steps) during the testing phase on the FIT dataset. We measure the performance change using SSIM, as presented in Figure 11. Several observations are highlighted: (1) As the gap between the training period and the testing time step increases, there is a general decline in model performance for all the methods. It can be seen that PENO$_{SA}$ has a relatively stable performance in long-term prediction, outperforming other methods in terms of accuracy. (2) The comparison amongst FNO, PRU, and PENO$_{SA}$ indicates that the integration of physical knowledge and the use of self-augmentation mechanisms in PENO$_{SA}$ effectively capture turbulence dynamics, which helps reduce accumulated errors in long-term simulations. (3) FNO struggles to achieve good performance starting from early testing phase.

**Visualization.** In Figure 12, the simulated flow data for the FIT dataset are displayed at multiple time steps (1st, 5th, 10th, and 20th) following the training period. For each time step, slices of the $w$

component at a specified $z$ value are presented. Several conclusions are highlighted: (1) At the 1st step, PENO$_{SA}$, PRU, FNO, and CTN obtain good performance because the test data closely resemble the training data at the last time step. In contrast, the baseline DSC/MS leads to poor performance starting from early time. (2) Beginning at the 5th time step, PENO$_{SA}$ starts to outperform FNO and PRU, with a more significant difference at the 20th time step. Specifically, FNO is unable to capture fine-level flow patterns due to the loss of high-frequency signals. While PRU is capable of capturing the complex transport patterns but introduces structural distortions and random artifacts due to accumulated errors in long-term simulations. In contrast, PENO$_{SA}$ addresses these issues effectively, resulting in significantly improved performance in long-term simulation.

**Ablation study for utilizing LES data.** This study aims to test the efficacy of incorporating LES into FNO and PENO$_{SA}$. The result of such integration is presented in Table 3, which indicates that both methods achieve improved accuracy when LES data is used to support flow data simulation. The flexibility in integrating LES is important as LES can often be generated at a low cost. It can also be seen that FNO's performance remains inferior to PENO$_{SA}$, even when FNO utilizes LES data and PENO$_{SA}$ does not utilize LES data. This observation also demonstrates the superiority of PENO$_{SA}$ method from another perspective.

## B.4 TRANSFERABILITY

To assess the transferability of PENO$_{SA}$, we evaluate the performance of PENO$_{SA}$, FNO, and PRU on the 2D vorticity dataset. Figure 13 shows the visual results at the 20th time step of the testing phase from the sequential test. It can be easily observed that PENO$_{SA}$ outperforms both FNO and PRU, capturing the flow patterns and magnitudes accurately. FNO fails to capture the correct patterns, and PRU can capture the flow patterns but has difficulty recovering the correct magnitudes of flow.

## B.5 SENSITIVITY ANALYSIS

Tables 4 and 5 provide the sensitivity analysis for parameter settings of random Gaussian perturbations (normal distribution) $\mathcal{N}$ from both the FIT and TGV datasets. Based on the results shown in tables, we can easily observe that the best parameter values for random Gaussian perturbations fall in the range of [0.1, 0.15].

