# OpenReview forum: "Physics-enhanced Neural Operator: An Application in Simulating Turbulent Transport"
_ICLR.cc/2025/Conference — Submitted to ICLR 2025_

### Official Review · Reviewer_gwQ8 · 2024-10-29

**Soundness:** 2
**Presentation:** 2
**Contribution:** 1
**Rating:** 3
**Confidence:** 5

**Summary:**

This paper proposes a physics enhanced neural operator (PENO) for regressing Direct Numerical Simulation (DNS) states to accurately  model turbulent flows. The proposed pipeline consists of a next-step neural predictor that takes an input state of the solver, consisting of a combination of DNS and Large Eddy Simulation (LES) variables (velocity/vorticity/pressure) at a given time step $t$. Then the PENO method predicts the next time step; this can be done recursively to predict a series of steps, effectively advancing the simulation without requiring a numerical solver. The proposed approach is based on Fourier Neural Operators (FNOs), which approximate the Navier-Stokes Partial Differential Equation (PDE) solution through a transformation of the input variables to Fourier space. However, trained FNOs with mean squared error losses have two shortcomings: limited generalization and reduced accuracy due ignorance of the underlying PDE, and impaired ability to capture high-frequency information of the learned dataset. The authors tackle these issues by estimating temporal gradients through direct assessment of the underlying PDE and through augmenting the regression with an additional network branch that performs super-resolution to capture high-frequency details.  The method is trained and tested in two datasets: the Forced Isotropic Turbulent flow (FIT) and the Taylor-Green vortex (TGV) flow, and the authors present some quantitative and qualitative analysis comparing the proposed approach against previous methods.

**Strengths:**

The authors correctly identify issues with the known  spectra bias in networks trained with MSE losses. These approaches tend to fail capturing crucial high-frequency information of the learned dataset. The proposed architecture is compared against previous baselines, and it shows interesting qualitative evaluations for next time step DNS prediction. Lastly, using the PDE information during training is an effective way to improve accuracy and generalization.

**Weaknesses:**

My biggest concern with papers that propose to replace solvers is their usual lack of fair comparison to a well implemented physics solver running on similar hardware (GPU) (for a thorough analysis check "Weak baselines and reporting biases lead to overoptimism in
machine learning for fluid-related partial differential equations"). A solver is a computational graph that can accurately predict the next state without the usual issues of data-driven approaches (lack of generalization, reduced accuracy). Without a clear intuition why a neural approach is able to make such a computational graph more efficient (e.g., reducing the dimensionality of the input through model reduction), it’s hard to justify why such a method is useful in the first place. This is the case for this paper, as the authors didn’t provide a fair comparison between their method and a DNS solver.  The authors could include a comparison of computational efficiency and accuracy between their method and a state-of-the-art DNS solver implemented on similar hardware for better assessing the efficiency of such proposed approach. Additionally, the authors could also more explicitly discuss the potential advantages of their neural approach over traditional solvers.

The paper also could have done a better job of evaluating errors in time, which are one of the common issues of regressing simulations with neural approaches, as this issue is not present in traditional solvers. Neural approaches that integrate in time tend to highly diverge relative to a ground truth solver, check “Stability analysis of chaotic systems from data” and “How Temporal Unrolling Supports Neural Physics Simulators” for further references. Furthermore, the authors could include a more detailed analysis of how prediction errors accumulate over time, showing more precise error metrics (apart from the ones presented in Figure 5) for different prediction horizons. You could also suggest they discuss how their method addresses (or doesn't address) the issue of divergence over time compared to traditional solvers.

Moreover, the chosen network architecture/approach for the regression of the next time step is outdated, since currently many other authors are now relying on the power of diffusion models to properly capture the complex behavior of fluid simulations. The authors even add a bit of noise to “improve generalization”, but it’s stated in the paper that a careful evaluation of such an approach is needed. I would suggest that just implementing a straightforward conditioned diffusion model would be a better solution. Otherwise, the authors could also compare their method against a state-of-the-art diffusion model approach, explaining why they chose their current architecture over more recent alternatives. The two-stage network solution also seems a bit cumbersome, and I suspect one could do the regression in a single-step. Perhaps the authors could do a better job at justifying why this approach was chosen over a single-step regression.

The manuscript, in its current version, also contains several exposition issues. Several typos (e.g, L025: “. we further”, L030: “results confirms”, L082: “training data are scarce”, L144: “denotes”, L174: “filtration” -> filtering, L223 (Figure 3): “Naiver Stoke”, L241: “appendix”, L275: “do not require”, L278: “, We create output”, L341: “5, 024”, L478: “. the simulated”, L494: “faces increased” are present. These errors reduce readability and can be confusing to the reader.

Lastly, the authors miss out several important references that are relevant to this work: The following papers are of direct relevance to the current submission: “Benchmarking Autoregressive Conditional Diffusion Models for Turbulent Flow Simulation”, “Uncertainty-aware Surrogate Models for Airfoil Flow Simulations with Denoising Diffusion Probabilistic Models”,  “Unsteady Cylinder Wakes from Arbitrary Bodies with Differentiable Physics-Assisted Neural Network” and “How Temporal Unrolling Supports Neural Physics Simulators”.

The aforementioned weaknesses and questions present in the next section justify my score for this paper.

**Questions:**

- At L141, it is included a transformation from the spatial domain to the Fourier domain for $\mathbf{Q}(t)$ at time $t$. Is the integration domain for the Fourier transformation to be a for a fixed time step, or is it for multiple time steps? The Fourier transform is written through integrating on $dt$, but I believe that the authors meant integration on the Fourier domain. Consider writing these variables differently for better exposition.

- At L209 the authors parametrize the non-linear function that represents the temporal derivative of $\mathbf{Q}$ through a parameter $\theta$. Why is this relevant? I could not find where this parameter was referenced in other places of the manuscript.

- Equation 1 denotes the evolution of an arbitrary variable $\mathbf{Q}$ when $\mathbf{Q} = \vec{v}$, where $\vec{v}$ is the fluid velocity. Why not write it only for velocity? If ones considers the Navier Stokes equation with regards to the vorticity, different formulations have to be considered (e.g., vorticity-streamfunction vs vorticity-velocity formulations), alongside with potential complications on the definition of boundary conditions.

- Why the authors didn’t include measuring the MSE in Tables 1 and 2? There seems to have space to do so, and it would benefit the quantitative analysis of the paper.

---

> ### Author Response · Authors · 2024-11-25
> **Thank you for your review.**
>
> We appreciate your time and efforts in providing insightful comments. Below, we tried our best to address your concerns.
>
> C1. At L141, it includes a transformation from the spatial domain to the Fourier domain for  at time . Is the integration domain for the Fourier transformation to be a for a fixed time step, or is it for multiple time steps? The Fourier transform is written through integrating on , but I believe that the authors meant integration on the Fourier domain. Consider writing these variables differently for better exposition.
>
> I acknowledge that it was our mistake to use the wrong symbol to represent tt. We will correct this in the final copyrighted version of the paper.
>
> C2. At L209 the authors parametrize the non-linear function that represents the temporal derivative of  through a parameter . Why is this relevant? I could not find where this parameter was referenced in other places of the manuscript.
>
> The symbol theta mentioned in line 209 does not represent the model's parameters. Instead, it is used as a general notation to describe the parameters contained in the Navier-Stokes (NS) equation.
>
> C3. Equation 1 denotes the evolution of an arbitrary variable  Why not write it only for velocity? If ones considers the Navier Stokes equation with regards to the vorticity, different formulations have to be considered (e.g., vorticity-streamfunction vs vorticity-velocity formulations), alongside with potential complications on the definition of boundary conditions.
>
> We aim to show  that the PDE-enhancement branch is not limited to the Navier-Stokes (NS) equation with respect to velocity but can also be applied to other variables. The only requirement is to adapt the branch to the corresponding PDE formulation for those variables.
>
> C4 Why the authors didn’t include measuring the MSE in Tables 1 and 2? There seems to have space to do so, and it would benefit the quantitative analysis of the paper.
>
> The evaluation metric, dissipation differences, serves a similar function to MSE. It calculates the pixel-wise difference between the generated DNS data and the real DNS data. Moreover, compared to MSE, dissipation differences can more effectively show the dynamic change between each pair of pixels within DNS flow data. More details on the explanation of dissipation differences can be found in the last paragraph of Section 4.1.
>
> To better clarification of the advantages of the proposed PENO method, we have provided a new quantitative analysis based on the MSE values in the FIT and datasets. I include my proposed PENO-based methods and selected baselines in these experiments. The results of FIT  are:
>
> DCS/MS     (0.158, 0.159, 0.158)
> FSR            (0.151, 0.153, 0.151)
> CTN            (0.123, 0.121, 0.123)
> FNO            (0.072, 0.071, 0.070)
> PRU            (0.065, 0.063, 0.063)
> PENO         (0.049, 0.048, 0.049)
> PENOSR    (0.038, 0.037, 0.038)
> PENOSA    (0.035, 0.034, 0.034)
>
> The results of TGV are:
>
> DCS/MS   (0.069, 0.070, 0.069)
> FSR          (0.062, 0.063. 0.062)
> CTN           (0.088, 0.089, 0. 088)
> FNO          (0.068, 0.069, 0.068)
> PRU           (0.039, 0.041, 0.040)
> PENO         (0.037, 0.038, 0.037)
> PENOSR    (0.029, 0.030, 0.029)
> PENOSA    (0.025, 0.026, 0.026)
>
> Based on the table, we can easily observe that the proposed PENO+ method can significantly outperform baselines, and the effectiveness of each proposed component in the proposed method can be justified by comparisons amongst FNO, PRU, and PENO-based methods.  In addition, we will include this supplementary experiment results in the appendix of the copyright version of the paper.

---

### Official Review · Reviewer_P9EC · 2024-10-30

**Soundness:** 2
**Presentation:** 2
**Contribution:** 2
**Rating:** 5
**Confidence:** 4

**Summary:**

This paper proposes to embed PDE-based constraints into FNOs for simulating fluids. Specifically, the paper targets turbulent transport, and long-term unrolling, i.e., many repeated invocations of the trained prediction network. FNOs have inherent known limitations in terms of the frequency spectrum that they can process. Here, the FNO blocks are augmented with “PDE branches” that predict the evolution the flow with a regular numerical method.

To prevent loosing smaller frequencies, the authors propose to predict two versions in parallel (low and high resolution), that are merged to produce an output that should better reflect the frequency distribution of the targets.

The method is evaluated on an isotropic turbulence case, and a taylor-green vortex. The authors also share their implementation at submission time, which is neat to see. Unfortunately, only very short rollouts of 10 to 20 steps are shown. For “real” applications in turbulence this seems very short.

Two recommendations regarding the presentation:

Fig. 4, does not really show the important part, the high frequencies. As often done in fluids papers, I can recommend rescaling by wavenumber (optionally with an exponent >1), or to provide some metrics on how much the frequency content improves.

Why give separate values for each channel in Table 1? You could simply mention that SRGAN has difficulties with the z-component, but it’s usually taken for granted that a “proper” architecture can handle multiple output channels.

**Strengths:**

The paper has several interesting aspects:
* It targets important and challenging scenarios
* Improvements over regular FNOs are demonstrated
* The submission comes with three-dimensional test cases
* A nice range of “pure prediction” baselines are included

**Weaknesses:**

One the negative side
* Effectively only 2 test scenarios are evaluated
* The metrics for evaluating the quality of the turbulent outputs are “unusual”, why not employ common metrics, starting with MSE, over energy spectra, TKE etc.?
* Section 3.2 is fairly unclear, Q-tilde and Q-hat are not used in any equations
* The length of the rollouts is extremely short. Long-term stability is not evaluated.

Maybe the largest problem I see with this submission is the omission of previous solver-in-the-loop approaches. These have a very similar goal to improve a coarse simulated baseline (here the evolution of eq. 1) with a learned model. The paper neither cites or discusses any of the related works, Um’2020, Kochkov’2021 etc. As far as I understand, the method in 3.1 corresponds exactly to a learned correction setup with an FNO as architecture. I still see merit in the subtler differences, and 3.2 seems to be non-standard (albeit a bit unclear), but these parallels should be made clear from the beginning. Similarly, the discussion of new aspects should put the work in context to those previous papers.

As many point were left open after the rebuttal and discussion, I don't recommend to accept the paper to ICLR in its current form. I think there are quite a few important open questions to address before publication.

**Questions:**

Please comment on similarities with solver-in-the-loop approaches, and clarify 3.2, as outlined above.

---

> ### Author Response · Authors · 2024-11-25
>
> We appreciate your time and efforts in providing insightful comments. Below, we tried our best to address your concerns.
>
> C1. The metrics for evaluating the quality of the turbulent outputs are “unusual”, why not employ common metrics, starting with MSE, over energy spectra, TKE etc.?
>
> The evaluation metric, dissipation differences, serves a similar purpose to MSE by calculating the pixel-wise difference between the generated DNS data and the real DNS data. However, compared to MSE, dissipation differences are more effective and sensitive in capturing the dynamic changes between each pair of pixels within the fine-grained DNS flow data. MSE, on the other hand, cannot adequately capture these dynamics. For example, as shown in FNO's results in the figure, the output flow data appears smooth, yet the MSE remains very low, failing to reflect the complex  variations in the flow.
> In addition, we will take your suggestions and include the figures of temporal or spatial spectrum of the flow-fields in the final copyrighted version of the paper.
>
> C2. Section 3.2 is fairly unclear, Q-tilde and Q-hat are not used in any equations
>
> Q-hat is defined on line 187 in Section 3.1, while Q-tilde is defined on line 283 in Section 3.2.
>
> C3. Please comment on similarities with solver-in-the-loop approaches
>
> The central idea of solver-in-the-loop approaches is to iteratively update the solver to enhance performance. However, our proposed method takes a different approach. FNO has significant limitations in simulating 3D turbulent flows, and even with iterative updates, it cannot achieve satisfactory simulation performance.
> Our method introduces a PDE-enhancement branch and a self-augmentation mechanism. These supplementary modules are specifically designed to overcome the weaknesses of FNO and significantly improve the overall performance of the model.

---

> > ### Comment · Reviewer_P9EC · 2024-11-26
> > **Rebuttal**
> >
> > I thank the authors for their updates and comments.
> >
> > To me it's still unclear where the difference to a solver-in-the-loop approach is. Doesn't the Kochkov et al. paper exactly target forced isotropic turbulence cases. The main difference seems replacing the CNN there, with an FNO here. That being said, the Kochkov CNN version seemed to be very stable, and produce rollouts for thousands of time steps.

---

### Official Review · Reviewer_iKBQ · 2024-10-31

**Soundness:** 3
**Presentation:** 3
**Contribution:** 2
**Rating:** 6
**Confidence:** 5

**Summary:**

Direct numerical simulations of turbulent flows can be prohibitively expensive to carry out. Fully data-driven or hybrid physics-based machine learning models can be quite promising in reducing the turnaround times for reconstructing fine scale data from coarse grained simulations or long time prediction of flow-field given historical DNS data. Motivated by the potential benefits of fourier neural operators in handling complex spatio-temporal data, the authors propose a physics enhanced neural operator method to model complex flow-field dynamics.
While FNOs work in a purely data driven fashion, the PENO in addition to data, also leverages the physics knowledge in the form of the underlying governing PDEs of turbulent flows. The authors also introduce a self-augmentation technique to enable long-time simulations/roll-outs. They demonstrate the model's capability on  different turbulent flow datasets and test across different resolutions as well.

**Strengths:**

The problem definition is clear with regard to the PENO being trained under a forecasting objective satisfying the physics constraints.
Instead of having a single network satisfy both data-driven and physics-based constraints, PENO has two branches, one FNO and the other physics based PDE branch. The final prediction / forecast is a weighted combination of the outputs of both branches.  Instead of using continuous derivatives, the authors use temporal and spatial discretisation to estimate the gradient terms in the governing equations. This is beneficial as it reduces the load on the neural network to strongly learn the continuous derivatives in the presence of only sparse data.
The authors clearly describe their methodology, datasets used, training and testing protocol.
They provide results validating their method across different benchmarks and other data-driven models.
The objective function is a simple forecasting MSE based loss function.This makes the learning easy and could prevent the competing objectives problem otherwise encountered in PINNs. Moreover, PENO allows multiple data sources to be combined as well such as DNS and LES through a weighted combination.

**Weaknesses:**

Although the authors claim novelty in the physics enhanced operator, the authors have seemed to ignore previous works on physics informed/enhanced operator learning in this domain. Physics informed neural operator: https://arxiv.org/pdf/2111.03794v3, Physics informed DeepONet https://ar5iv.labs.arxiv.org/html/2207.05748 and its derivatives. Instead of comparing their model with other operator learning frameworks, they infact compare with some of the super-resolution models which in some way seems out of context. It would have been better if the authors could have provided comparison with the other previously proposed Physics based operator learning frameworks which came out much earlier than this work. While the contours look decent in the results, comparison of spatial and temporal spectrum would be worthwhile in showing if the model is capable of overcoming spectral bias that otherwise plagues neural networks in general. It is not clear if PENO can operate in a purely physics based training regime without the data-driven component as in PINO.

**Questions:**

1. Could you elaborate on the key differences between PENO and Physics informed neural Operator?
2. Is there any reason to not compare your model with other operator learning based frameworks and not include them in the survey or in this study?
3. Could you explain why temporal or spatial spectrum of the flow-fields have not been included in the evaluation? Any justification for why dissipation different has been considered?
4. Can PENO be used in a fully data-free regime as in the PINO paper? Where given only the initial and boundary conditions, can the model be trained to simulate turbulent flow?

---

> ### Author Response · Authors · 2024-11-25
> **Thank you for your review.**
>
> We appreciate your time and efforts in providing insightful comments. Below, we tried our best to address your concerns.
>
> C1. Could you elaborate on the key differences between PENO and Physics informed neural Operator?
>
> The major difference between our proposed PENO and PINO lies in how they incorporate the FNO benchmark and utilize PDE information. PINO uses PDEs as a physics-informed loss to regularize the model during training. In contrast, our proposed PENO integrates a PDE-enhancement branch directly into the model architecture, embedding the PDE format into the structure itself. This structural integration allows PENO to inherently encode the physical principles, rather than relying solely on loss-based regularization.
> Incorporating PDEs directly into the model structure, rather than as a loss function, offers significant advantages grounded in KGML theory. By embedding PDEs into the model architecture, the physical laws become an integral part of the representation, ensuring that the model inherently adheres to these principles throughout training and inference. This structural integration reduces the reliance on large datasets and mitigates the risk of overfitting to noise, as the encoded physics acts as a strong prior. In contrast, using PDEs as a loss function primarily guides the optimization process but does not guarantee strict adherence to physical laws in the learned representations, it is more unstable.
>
> C2  Is there any reason to not compare your model with other operator learning based frameworks and not include them in the survey or in this study?
>
> We provide the following supplementary experimental results (left: SSIM, right:  Dissipation diff) on both the FIT and TGV datasets for your reference. We will also include these results in the final copyrighted version of the paper.
>
> FIT dataset:
> DeepOnet (0.909, 0.911, 0.909)  (0.156, 0.154, 0.155),
> FNO (0.912, 0.915, 0.911)  (0.153, 0.151, 0.150),
> PiDeepOnet (0.923, 0.923, 0.924) (0.143, 0.142, 0.142),
> PINO (0.931, 0.933, 0.934) (0.134, 0.133 0.133),
> PENO (0.968, 0.972, 0.967)  (0.110, 0.107, 0.110)
>
> TGV dataset
> DeepOnet (0.641, 0.638, 0.642)  (0.075, 0.074, 0.074),
> FNO (0.645, 0.646, 0.648)  (0.072, 0.071, 0.072),
> PiDeepOnet (0.705, 0.706, 0.705) (0.043, 0.042, 0.043),
> PINO (0.723, 0.722, 0.721) (0.040, 0.042, 0.040),
> PENO (0.843, 0.847, 0.844)  (0.032, 0.033, 0.034)
>
> From the above results, we can easily find that our proposed PENO outperforms all of these neural operator based methods.
>
> C3. Could you explain why temporal or spatial spectrum of the flow-fields have not been included in the evaluation? Any justification for why dissipation different has been considered?
>
> We will take your suggestions and include the figures of temporal or spatial spectrum of the flow-fields  in the final copyrighted version of the paper.
>
> C4. Can PENO be used in a fully data-free regime as in the PINO paper? Where given only the initial and boundary conditions, can the model be trained to simulate turbulent flow?
>
> Yes, this is included in the transferability experimental results shown in Figure 8 and Section 4.3 on page 10.

---

> > ### Comment · Reviewer_iKBQ · 2024-11-25
> > **Acknowledgement of author response and additional comments.**
> >
> > Dear Authors,
> >
> > Thanks for addressing my questions.
> >
> > **Comparison with other operator learning frameworks**
> > 1. The performance comparison of the operator learning methods on FIT and TGV datasets indeed indicate that PENO outperforms the rest.
> > 2. It would also be better to bring forth the key features/differences between these methods and PENO in the literature/ methodology section to highlight your contributions better.
> > 3. Spatial/spectrum plots would highlight the reconstruction performance of the model better.
> >
> > **Transferrability/ forward problems**
> > 1. If I understand corectly, the transferrability experiment is carried out over a trained model but with or without fine tuning for different PDEs than the one used to train the PENO in the first place.
> > 2. Training of PENO still requires data.
> > 3. However, I would be interested to know whether given the PDE structure in the PENO architecture, is it possible to train PENO to solve a forward problem with just initial and boundary conditions available while training. Basically, can PENO function as a CFD solver like as Deep Galerkin Methods/ Deep Ritz Methods/PINNs?
> >
> > Thanks

---

### Official Review · Reviewer_yHQa · 2024-11-02

**Soundness:** 2
**Presentation:** 1
**Contribution:** 3
**Rating:** 5
**Confidence:** 3

**Summary:**

The authors provide an interesting approach to Physics-informed ML. However I cannot recommend this paper for acceptance unless (i) the paper can clarify some missing information  and (ii) additional information is provded on the model comparison with other approaches (see questions). This is nonetheless promising work -- hopefully the authors can address these shortcomings in paper presentation in the rebuttal.

**Strengths:**

1. Good motivation for neural operator work -- surrogates for DNS are a major research focus.
2. Good test cases GIT, FIT and Taylor-Green are well accepted configs. in the physics community.
3. Potentially promising results.

**Weaknesses:**

1. Quantitative evaluation in table 2 is missing model parameters/FLOPs to discount scaling effects on architecture comparison.
2. More information on training data could be provided in appendix. See question 4.

**Questions:**

1. Line 76 -- Are Neural Operators really that efficient compared to other deep learning approaches? Chung et al (NeurIPS 2023) showed that Fourier neural operators in turbulent flow applications can have large matrix dimensionalities that result in poor scaling behavior when compared to other NN architectures.
2. Figure 2 -- How does it make sense to have both DNS and LES data at time t as inputs? After an initial condition, LES and DNS behavior would drift from each other since the missing physics in LES would change the trajectory of the simulation. There is an ablation study to show minor improves from LES inputs. But some clarification on fundamental mechanism would be useful for readers.
3. Table 2 -- what's the number of parameters/FLOPs of the different architectures. Model scales can affect predictive performance -- see Chung et al (NeurIPS 2023).
4. What numerical solver for data? What is the spatial and temporal differencing scheme? Are flow compressible or incompressible? What is the resolution of DNS w.r.t Kolmogorov lengthscales?
5. What are the exact numbers of training and test samples used for each case? A table in Appendix could be useful for clarity.

---

> ### Author Response · Authors · 2024-11-24
> **Thank you for your review.**
>
> We appreciate your time and efforts in providing insightful comments. Below, we tried our best to address your concerns.
>
> C1. Quantitative evaluation in table 2 is missing model parameters/FLOPs to discount scaling effects on architecture comparison.
>
> The total parameters of the models are as follows: FNO (62,337), PRU (84,938), and PENO (151,284). SR methods typically have over a million parameters. Compared to SR-based methods, PENO remains efficient.
>
> C2. Line 76 -- Are Neural Operators really that efficient compared to other deep learning approaches? Chung et al (NeurIPS 2023) showed that Fourier neural operators in turbulent flow applications can have large matrix dimensionalities that result in poor scaling behavior when compared to other NN architectures.
>
> The FNO requires less than 5 minutes for training and less than 10 seconds for inference. The PRU takes approximately 8 minutes for training and about 20 seconds for inference. PENO, on the other hand, takes around 10 minutes for training and about 30 seconds for inference. In contrast, other neural network architectures often require over an hour for training and several minutes for inference. This makes PENO significantly more efficient in terms of both training and inference time compared to other neural network architectures.
>
> C3. Figure 2 -- How does it make sense to have both DNS and LES data at time t as inputs? After an initial condition, LES and DNS
> behavior would drift from each other since the missing physics in LES would change the trajectory of the simulation. There is an ablation study to show minor improves from LES inputs. But some clarification on fundamental mechanism would be useful for readers.
>
> In data-driven models, such as the one shown in Figure 2, error and bias can accumulate over time, especially in the later stages of simulation. While data-driven methods like FNO can effectively simulate data in the short term, their performance tends to degrade over longer time horizons as errors compound. On the other hand, although LES data has inherent biases due to missing physics, it serves as a valuable supplement for data simulation, particularly in longer-term scenarios. As demonstrated in Table 3 of the appendix, incorporating LES data can lead to notable improvements, especially during the later stages of the simulation. This combined approach leverages the strengths of both data-driven methods and LES to enhance overall accuracy of data simulation.
>
> C4. What numerical solver for data? What is the spatial and temporal differencing scheme? Are flow compressible or incompressible? What is the resolution of DNS w.r.t Kolmogorov lengthscales?
>
> Both FIT and TGV are incompressible. The numerical solver used is a pseudo-spectral method. The spatial and temporal information is provided in Section B.1 of the dataset description.
>
> C5. What are the exact numbers of training and test samples used for each case? A table in Appendix could be useful for clarity.
>
> The number of training and testing samples is described in the experimental design section on page 7. I will incorporate your suggestion to summarize this information in a table in the final copyrighted version of the paper.

---

> > ### Comment · Reviewer_yHQa · 2024-11-25
> >
> > Thank you for addressing some of my concerns.
> >
> > 1. and 2. Thanks for doing the param calculations, but what are the  theoretical FLOPs values of the different architectures studied in this paper. A clearer presentation of this can help readers.
> >
> > 3. This is sufficient discussion. Thank you. Please add this to revised paper, if not already present.,
> >
> > 4. Kolmogorov lengthscales are still not mentioned.
> >
> > 5. Number of train and test samples are still not explicitly mentioned on pg 7.

---

### Official Review · Reviewer_cPpW · 2024-11-03

**Soundness:** 3
**Presentation:** 3
**Contribution:** 2
**Rating:** 5
**Confidence:** 4

**Summary:**

The paper presents a Physics-Enhanced Neural Operator (PENO) model designed to improve the simulation of turbulent flows. PENO integrates physical knowledge of partial differential equations (PDEs) with a Fourier Neural Operator (FNO) framework, aiming to capture complex flow dynamics accurately. The model introduces a self-augmentation mechanism to maintain high-frequency information, which is often lost in traditional FNO models. Evaluations on multiple turbulent flow datasets show that PENO effectively reconstructs high-resolution DNS data, generalizes across resolutions, and mitigates error accumulation in long-term simulations. This demonstrates PENO's applicability to various scientific and engineering fields where extensive, high-resolution simulations are essential but computationally expensive.

**Strengths:**

+ Integrates physics-based knowledge directly into the neural operator framework, enhancing the ability to capture complex flow patterns that purely data-driven models often miss.
+ Demonstrates high generalizability across various datasets and spatial resolutions, showing adaptability to different turbulent flow conditions.
+ Introduces a self-augmentation mechanism that preserves high-frequency flow patterns over time, essential for maintaining accuracy in simulations that run for extended periods.
+ Outperforms existing models in structural similarity index measure (SSIM) and dissipation difference metrics, indicating superior performance in both spatial accuracy and gradient capturing ability.

**Weaknesses:**

- Relies on large eddy simulation (LES) data for optimal results, which, while cheaper than DNS data, still adds to the data requirements and may not always be available.
- Risk of overfitting in long-term simulations, particularly when adjusting random Gaussian perturbation parameters, as extensive tuning is needed to balance accuracy and robustness.
- Challenges with maintaining accuracy in extremely high-dimensional or fine-grained simulations, where capturing all necessary dynamics requires considerable computational power and data.
- Not sure if the pressure predictions also are of high quality.

**Questions:**

1. What about the error in the predicted pressures? It has been known that it is more dificult to predict the pressures using neural operators.
2. How easy would it be to extend the approach to non rectilinear domains. One of the main limitations of FNO is that they are suitable only for rectilinear domains.

---

> ### Author Response · Authors · 2024-11-24
> **Thank you for your review.**
>
> We appreciate your time and efforts in providing insightful comments. Below, we tried our best to address your concerns.
>
> C1 Risk of overfitting in long-term simulations, particularly when adjusting random Gaussian perturbation parameters, as extensive tuning is needed to balance accuracy and robustness.
>
> I don't think PENO requires extensive additional tuning. The reasons are as follows: (1) As shown in Tables 1 and 2, when comparing PENO, PENO_SR (without Gaussian perturbation), and PENO_SA, the upscaling step provides a significantly greater improvement than the Gaussian perturbation. (2) As shown in Table 5 in the appendix, varying the parameters of the Gaussian perturbation only causes the performance to fluctuate within a very narrow range. This indicates that the model's sensitivity to these parameters is minimal, reducing the need for extensive tuning.
>
> C2 Challenges with maintaining accuracy in extremely high-dimensional or fine-grained simulations, where capturing all necessary dynamics requires considerable computational power and data.
>
> I agree with your comments that achieving accuracy in such simulations does require additional computational power and data. However, our primary goal is sequential prediction rather than super-resolution. We assume that a portion of high-resolution data is available at the beginning of the simulation to train the model for capturing the physical pattern. Using this initial data, the model can effectively generate predictions for future time periods. Additionally, we have the option to incorporate LES data to support the simulation process, which can help achieve better results.
>
> C3 What about the error in the predicted pressures? It has been known that it is more difficult to predict the pressures using neural operators.
>
> We use pressure values solely to support the generation of velocity data in the PDE-enhancement branch. In our setup, we do not predict pressure values.
>
> C4 How easy would it be to extend the approach to non rectilinear domains? One of the main limitations of FNO is that they are suitable only for rectilinear domains.
>
> PENO can extend to non-rectilinear domains. One potential approach involves coordinate mapping, where the irregular domain is transformed into a rectilinear grid using a mapping function (e.g., curvilinear coordinates). The PENO operates in the transformed space, and the results are mapped back to the original domain. Another approach replaces the global Fourier transform with pseudo-spectral methods that use localized basis functions, such as Chebyshev polynomials or wavelets, to approximate the spectral representation on irregular grids.

---

### Official Review · Reviewer_c3wa · 2024-11-04

**Soundness:** 4
**Presentation:** 4
**Contribution:** 3
**Rating:** 6
**Confidence:** 3

**Summary:**

This paper is focused on improving the Fourier Neural Operator, incorporating physical knowledge and making some architectural improvements in order to better simulate turbulent flow.  By adding physics knowledge, adding an extra network branch, and adding high-frequency signals at each time step, the method is designed to outperform existing techniques for this application.  The authors' PENO method is evaluated on well-known datasets / test problems.

**Strengths:**

1. The paper is written well, and the figure quality is high.

2. The paper makes several novel (from what I can tell) modifications to the existing FNO algorithm, and it appears that these all turn out to be improvements.

3. Results seem quite positive: it looks like the PENO method outperforms the target baselines exactly as hoped by the authors.

**Weaknesses:**

1. It is not clear how much each of the various improvements contributes to the overall success of the method.  For example, if you just use FNO with the additional self-augmentation mechanism or just with the additional physics data, how well do things perform compared to what is ultimately the PENO method?

2. Fig. 6 is somewhat helpful, but it might be more informative to instead show the DNS result, and then just plot the errors of |method - DNS| for each method?  In particular, since you're using SSIM, readers may naturally want to look at the spatial distributions of errors of the various methods, not just the magnitudes, so that could be helpful.

3. I think there are some missing details around training: what optimization algorithm was used, what learning rates, how long did training take (and ideally how does this compare to other methods like FNO), what do the training and validation loss curves look like (so we can see how much overfitting may be happening), etc.

Minor:

"In the field of computational fluid dynamics (CFD)," just say "In CFD," here

**Questions:**

1. Table 1: Why were SSIM and dissipation difference the chosen metrics here?  This could be informative for readers to understand.

1. Could the authors speak at all to the generalizability of the PENO technique to problems besides turbulent flow?  How do the authors imagine other researchers building upon and extending the present work?

---

> ### Author Response · Authors · 2024-11-24
> **Thank you for your reviews.**
>
> We appreciate your time and efforts in providing insightful comments. Below, we tried our best to address your concerns.
>
> C1: It is not clear how much each of the various improvements contributes to the overall success of the method. For example, if you just use FNO with the additional self-augmentation mechanism or just with the additional physics data, how well do things perform compared to what is ultimately the PENO method?
>
> I conducted new experiments with the FNO incorporating an additional self-augmentation mechanism. The quantitative performance metrics are as follows: SSIM (0.767, 0.761, 0.763) and Dissipation Difference (0.038, 0.039, 0.038). These results are worse than those achieved with the proposed PENO method.
> In Table 3 of the appendix, we provide the experimental results for FNO with additional LES data. The performance is worse compared to PENO without LES data.
>
> C2: Fig. 6 is somewhat helpful, but it might be more informative to instead show the DNS result, and then just plot the errors of |method - DNS| for each method? In particular, since you're using SSIM, readers may naturally want to look at the spatial distributions of errors of the various methods, not just the magnitudes, so that could be helpful.
>
> We will take your suggestions and include the figures of spatial distribution in the final copyrighted version of the paper.
>
> C3: I think there are some missing details around training: what optimization algorithm was used, what learning rates, how long did training take (and ideally how does this compare to other methods like FNO), what do the training and validation loss curves look like (so we can see how much overfitting may be happening), etc.
>
> The implementation details are provided in appendix B.2.
>
> C4: Table 1: Why were SSIM and dissipation difference the chosen metrics here? This could be informative for readers to understand.
>
> The reason we selected SSIM is that it is highly sensitive to structural information, making it a suitable metric for evaluating the overall structural quality of generated data. Additionally, in computational fluid dynamics, researchers not only focus on the overall structure but also pay close attention to spatial variations in detail. Therefore, we use Dissipation Difference, a commonly used metric in this field, as another standard.
>
> C5: Could the authors speak at all to the generalizability of the PENO technique to problems besides turbulent flow? How do the authors imagine other researchers building upon and extending the present work?
>
> PENO is fundamentally a Partial Differential Equation (PDE) solver, designed to handle systems governed by PDEs. While this work focuses on turbulent flow—a highly complex and challenging case in fluid dynamics—PENO's applicability is not restricted to this specific type of flow. It can also be effectively applied to simulate other types of fluid flows, such as water flow, atmospheric flow, and other scenarios commonly encountered in fluid dynamics.
>
> The versatility of PENO extends beyond fluid dynamics. As a general PDE solver, it has the potential to be utilized in a wide range of dynamical systems that are guided by PDEs. These systems could include, but are not limited to, heat transfer, electromagnetism, and chemical reaction-diffusion processes. By adapting the solver to the specific equations and physical constraints of a given system, researchers can leverage PENO to address challenges in various scientific and engineering domains.

---

> > ### Comment · Reviewer_c3wa · 2024-11-24
> >
> > Thanks for engaging with the reviewers.  I believe that incorporating points like your responses to C4 and C5, and including the new (or newly proposed) results regarding C1 and C2 into the paper, will be quality improvements for the manuscript.  These aren't major changes (I had expected the authors would be able to address them), and so in my view, my score will remain the same.  I will monitor other changes made and discussions with other reviewers to see if further reevaluation is warranted.

---

### Comment · Area_Chair_khZe · 2024-11-25
**Reviewers' Response**

Dear Reviewers,

As the author-reviewer discussion period is approaching its end, I would strongly encourage you to read the authors' responses and acknowledge them, while also checking if your questions/concerns have been appropriately addressed.

This is a crucial step, as it ensures that both reviewers and authors are on the same page, and it also helps us to put your recommendation in perspective.

Thank you again for your time and expertise.

Best,

AC

---

### Meta-Review · Area_Chair_khZe · 2024-12-18

**Metareview:**

This paper introduces the Physics-Enhanced Neural Operator (PENO), a novel approach that builds upon Fourier Neural Operators (FNOs) to emulate turbulent flows. PENO aims to overcome the limitations of traditional FNOs, including limited generalization, reduced accuracy from neglecting underlying physics, and the inability to capture high-frequency details. PENO attempts to improve upon standard FNOs by incorporating physical knowledge of the underlying PDEs. This is done through a self-augmentation mechanism that the authors claim it retains high-frequency information by integrating "PDE-enhancement branches" to predict the flow evolution using numerical methods plus a  super-resolution which is then weighted with a regular FNO branch. The method is evaluated on two datasets: Forced Isotropic Turbulence (FIT) and Taylor-Green Vortex (TGV) flow.

Reviewers found the approach promising but raised several concerns. These included the limited length of the demonstrated rollouts, the need for clarification on certain aspects of the method, and comparisons with other approaches, as some important related literature seemed to be absent. Concerns were also raised regarding the rationale for using both LES and DNS inputs, which can diverge due to the chaotic nature of the problem. Additionally, the metrics used are not standard in CFD, making it difficult to gauge the method's behavior and advantages. Several ablation studies are missing, making the relative importance of each innovation to the overall performance unclear. The authors provided only a brief response during the rebuttal period, failing to address the reviewers' concerns or provide an updated manuscript. As such, I recommend rejection.

**Additional Comments On Reviewer Discussion:**

Some of the reviewers, raised the issue that the paper seems oblivious to a large portion of literature, that the benchmarks are lacking, and the evaluation metrics are usually for computer vision and not for the target domain. The authors provided a succinct response but provided no modifications to the manuscript.

---

### Decision · Program_Chairs · 2025-01-22

Reject